# ASCENSION: Autoencoder-Based Latent Space Class Expansion for Time Series Data Augmentation

## Abstract

Achieving effective data augmentation (DA) in time series classification is challenging due to the diverse nature of temporal data. While state-of-the-art generative models for DA – *based on GANs, diffusion models, or Variational Autoencoders (VAEs)* – demonstrate potential, they often fail to deliver consistent improvements across various datasets and application domains (e.g., ECG, power consumption, vibration sensor data), as confirmed in this study. To address this limitation, we introduce ASCENSION (**A**utoencoder-based latent **s**pace **c**lass expa**nsion**), a novel generative approach that harnesses the probabilistic structure of the VAE's latent space alongside an innovative controlled and progressive class expansion mechanism. It promotes compact intra-class representations while maximizing inter-class separability, thereby reducing the likelihood of class overlap during latent space exploration. We evaluate ASCENSION on 102 datasets from the UCR benchmark and compare it against six state-of-the-art DA methods. Empirical results show that ASCENSION improves average classification accuracy by approximately $1\%$, whereas the strongest competing method leads to an average accuracy change of $-0.3\%$. ASCENSION achieves a non-negative improvement in classifier performance for $66.2\%$ of the 102 datasets — a $16.4\%$ improvement over the previous best method. These results establish ASCENSION as the first DA method that can be reliably applied in real-world scenarios where prior knowledge of method suitability is uncertain. Our study further explores the key factors driving its superior performance.

## 1. Introduction

Time series classification (TSC) is challenging due to temporal dependencies, non-stationarity, and limited labeled data. Real-world constraints, such as high collection costs and privacy regulations, further restrict training set sizes and impact model accuracy. Data augmentation (DA) helps mitigate these constraints by generating synthetic samples that increase both the quantity and diversity of training data. Formally, given a labeled dataset $\{x_i^y\}$ for each class $y \in \{1, 2, \ldots, Y\}$, DA aims to create additional synthetic samples that preserve class semantics while broadening coverage of the data distribution.

DA methods generally fall into two categories: *traditional* and *generative* (Iglesias et al., 2023b). Traditional methods such as AutoAugment (Cubuk et al., 2019) and Fast AutoAugment (Lim et al., 2019) apply predefined transformations (e.g., jittering, window slicing, scaling). While effective in image classification, their application to time series is often hindered by the risk of disrupting crucial temporal patterns, such as periodicity or phase alignment.

*Generative DA* methods, based on GANs, diffusion models, and VAEs (Cheung & Yeung, 2020), bypass such handcrafted transformations by learning to model the underlying data distribution. GAN-based methods, such as TimeGAN (Zhang et al., 2022), TTS-GAN, LatentAugment (Tronchin et al., 2023), can produce high-quality, rapidly sampled time-series, but may exhibit limited diversity (Xiao et al.). Diffusion models generate rich, varied samples at the cost of high computational overhead (Feng et al., 2024). VAE-based methods often strike a promising balance, providing relatively fast sampling within a structured latent space, but offer limited means to *expand* beyond the distribution already seen in the training data.

To our knowledge, no state-of-the-art DA method for time-series classification enables progressive (iterative) and meaningful class boundary expansion during synthetic data generation. This limitation, discussed further in Appendix A and Figure 6, becomes critical when training and operational data distributions diverge (i.e., distribution discrepancy ratio), often due to factors like sensor drift, unseen conditions, or temporal shifts. To bridge this gap, we propose **ASCENSION**, a novel VAE-based DA framework that preserves fast sampling and flexible latent representations while enabling controllable class boundary extrapolation. Unlike conventional generative DA methods that strictly replicate the training set's latent distribution, ASCENSION features a tunable mechanism for exploring underrepresented or unseen regions without intruding into overlapping or ambiguous

Figure 1: Visualization of the latent space dynamics in traditional generative DA methods versus ASCENSION. Traditional methods sample new points within the learned data distribution, limiting diversity and restricting class representation. In contrast, ASCENSION incorporates a controllable and progressive boundary expansion mechanism, enhancing inter-class separation to generate extrapolated yet class-consistent synthetic samples, allowing for more flexible and representative DA.

class areas. Specifically, it leverages the probabilistic structure of the VAE's latent space through a multi-component representation per class. By adjusting these components, ASCENSION enables controlled and progressive expansion of class probability densities and boundaries. Additionally, ASCENSION enforces structural constraints that ensure intra-class compactness while maintaining inter-class separation, preserving class consistency and preventing harmful overlap. This leads to richer, more representative synthetic time-series data, enhancing diversity and ultimately improving classification performance. To highlight ASCENSION's originality compared to existing generative DA methods, Figure 1 illustrates its latent space dynamics versus traditional generative DA methods.

Our key contributions are:

1. **Novel VAE-based DA Method:** ASCENSION pioneers a controllable and progressive boundary-expansion mechanism, unlocking richer generative spaces and significantly enhancing applicability against distribution discrepancies, a crucial challenge in real-world TSC applications;

2. **Unparalleled Benchmarking & Performance Gains:** We rigorously evaluate ASCENSION's impact on classification performance across a vast and diverse set of time-series datasets, outperforming both traditional (FAA) and generative methods (LatentAugment, TTS-GAN, Time-DDPM, VaDE, and MODALS);

3. **Fundamental Data-driven Insights:** We analyze how different time-series properties influence DA performance, showing that ASCENSION's controlled extrapolation can better align training and operational distributions.

The rest of the paper is structured as follows. Section 2 discusses related DA methods, covering both traditional and generative methods. Section 3 presents the ASCENSION framework. Section 4 then provides an extensive empirical evaluation and comparative analysis. Finally, we conclude with key takeaways and future directions.

## 2. Related Work

DA for time series falls into traditional and generative methods. Traditional methods like window slicing, jittering, and scaling (Iglesias et al., 2023a) apply transformations from computer vision but often distort temporal and semantic integrity. Automated methods such as AutoAugment (AA) (Cubuk et al., 2019) optimize transformations via reinforcement learning, while Fast AutoAugment (FAA) (Lim et al., 2019) improves efficiency with density matching. Further refinements, including RandAugment (Cubuk et al., 2020), Deep AutoAugment (Zheng et al., 2022), and Trivial Augment (Müller & Hutter, 2021), streamline augmentation strategies. However, these methods still rely on predefined transformations, limiting adaptability to complex time series.

Generative DA methods, leveraging models like GANs, VAEs, and diffusion models, offer more flexible augmentation by learning probabilistic representations of time series distributions. TimeGAN (Zhang et al., 2022), TS-GAN(Yang et al., 2023b), and TTS-GAN(Li et al., 2022) adapt GAN architectures for time series, capturing long-range dependencies and improving data quality. However, GANs suffer from training instability, sensitivity to hyper-parameters, and mode collapse. More recent advances in diffusion models, such as ASE-DDPM (Liu et al., 2024), DiffRUL (Wang et al., 2024), and Time-DDPM (Solis-

Martin et al., 2023), have demonstrated improved stability but struggle with long-range dependencies and slow inference. VAEs, by contrast, provide a more structured latent space, facilitating better sample diversity control. MODALS (Cheung & Yeung, 2020) was the first VAE-based approach to explore class boundary expansion, though without a controllable mechanism. VAE-LSTM (Dang et al., 2024) and VaDE (Jiang et al., 2016) have also been proposed for time series augmentation but do not explicitly model class expansion, a gap addressed by ASCENSION.

For a more detailed discussion on "Related Work", refer to Appendix A, which also highlights the state-of-the-art methods benchmarked in this study, as summarized in Figure 6.

## 3. ASCENSION framework

Unlike traditional generative DA methods that apply input-space transformations (e.g., random warping or scaling), which can lead to sample degradation or unintended class confusion, ASCENSION explicitly models class-conditional densities and incorporates a *risk-aware exploration mechanism*, regulated by a scaling factor $\alpha$, to mitigate class overlap and ensure high-quality augmentations. ASCENSION is designed to achieve a delicate balance between three objectives: (1) precise VAE-based density modeling; (2) risk-aware exploration to prevent degenerate samples, and (3) controlled class distribution expansion, enabling diverse and useful synthetic data for time series classification.

Sections 3.1 and 3.2 detail how ASCENSION integrates VAE training and clustering constraints respectively. Section 3.3 details the proposed iterative class expansion mechanism expanding these latent distributions iteratively to produce synthetic data.

### 3.1. VAE Training & Latent Space

ASCENSION begins with a VAE that models data $\mathbf{X}$ in a probabilistic latent space. We optimize the Evidence Lower Bound (ELBO),

$$\mathcal{L}(\theta, \phi) = \mathbb{E}_{q_\phi(\mathbf{z}|\mathbf{x})}\left[\log p_\theta(\mathbf{x}|\mathbf{z})\right] - D_{KL}\Big(q_\phi(\mathbf{z}|\mathbf{x}) \,\|\, p(\mathbf{z})\Big), \quad (1)$$

where $q_\phi(\mathbf{z}|\mathbf{x})$ is the approximate posterior, $p_\theta(\mathbf{x}|\mathbf{z})$ is the likelihood, and $D_{KL}$ is the Kullback-Leibler divergence from the prior $p(\mathbf{z})$. To capture class-specific nuances, ASCENSION estimates each class's distribution in the latent space, enabling controlled sampling and mitigating ambiguity among overlapping regions.

### 3.2. Clustering Constraints

To enhance class separability, ASCENSION incorporates a clustering loss:

$$\mathcal{L}_{\text{cluster}} = \sum_{i=1}^{N} \sum_{j=1}^{N} \delta_{y_i, y_j}\, d(\mathbf{z}_i, \mathbf{z}_j), \quad (2)$$

where $\delta_{y_i, y_j} = 1$ if samples $i$ and $j$ share the same class, and 0 otherwise; $d(\mathbf{z}_i, \mathbf{z}_j)$ is the distance metric (cosine similarity). This loss function reinforces *intra-class compactness* while maximizing *inter-class separability*, ensuring well-structured latent clusters for generating more consistent and reliable synthetic samples.

### 3.3. Latent Class Expansion

ASCENSION iteratively expands each class's latent distribution following a five-step process:

1. **Train the VAE with Clustering**:

$$\mathcal{L}_{\text{VAE}} = \mathcal{L}_{\text{recon}} + \mathcal{L}_{\text{KL}} + \mathcal{L}_{\text{cluster}} + \mathcal{L}_{\text{class}},$$

   optimized over the current training set;

2. **Sample Latent Points**: For each class $y$, sample new points from a Gaussian mixture centered on class-specific means:

$$\frac{1}{K_y} \sum_{k=1}^{K_y} \mathcal{N}\big(\mu_{y,k}, \alpha\, \Sigma_{y,k}\big), \quad (3)$$

   where $\alpha$ scales the covariance to systematically *expand* the class boundaries;

3. **Label Assignment via Posterior Probability**: If sampled points lie in overlap regions, assign labels by maximizing the posterior probability to ensure risk-aware augmentation and avoid misclassification;

4. **Decode and Augment**: Decode latent points into time series, then add them (with labels) to the training dataset, enriching its variety without jeopardizing class integrity;

5. **Retrain Iteratively**: Use the augmented dataset to retrain the model from scratch, refining its parameters and further exploring latent regions over multiple iterations.

This five-step process is formalized in Algorithm 1. Empirical results (Section 4 and Appendix B) show that values of $\alpha$ slightly above 1 effectively boost diversity without sacrificing class consistency.

**Algorithm 1** Augmentation Loop with distinct classes

1: **Input:** Original time series data $\mathbf{X} = \{\mathbf{x}_1, \mathbf{x}_2, \ldots, \mathbf{x}_n\}$ with class labels $\mathbf{Y} = \{y_1, y_2, \ldots, y_n\}$
2: **Output:** Augmented training dataset $\mathbf{X}_{\text{aug}}, \mathbf{Y}_{\text{aug}}$
3: **Initialization:**
4: $\mathbf{X}_{\text{aug}} \leftarrow \mathbf{X}$
5: $\mathbf{Y}_{\text{aug}} \leftarrow \mathbf{Y}$
6: **while** augmentation desired **do**
7:    **Train VAE:**
8:    $\mathcal{L}_{\text{VAE}} = \mathcal{L}_{\text{recon}} + \mathcal{L}_{\text{KL}} + \mathcal{L}_{\text{cluster}} + \mathcal{L}_{\text{class}}$
9:    $\theta^*, \phi^* \leftarrow \arg\min_{\theta,\phi} \mathcal{L}_{\text{VAE}}$ using $\mathbf{X}, \mathbf{Y}$
10:    **Build combination of Gaussian:**
11:    Let $d_y = \frac{1}{K_y} \sum_{k=1}^{K_y} \mathcal{N}(\mu_{y,k}, \alpha\Sigma_{y,k})$ to $\mathbf{Z}$ for each class $y$
12:    **Sample Latent Points:**
13:    **for** each class $y$ **do**
14:      $\mathbf{Z}_{\text{new}}^y = \{\mathbf{z}_1^{ly}, \mathbf{z}_2^{ly}, \ldots, \mathbf{z}_m^{ly}\} \sim d_y$
15:      **for** each $\mathbf{z}_i^{ly} \in \mathbf{Z}_{\text{new}}^y$ **do**
16:        If $\mathbf{z}_i^{ly}$ has higher probability of being in class $y'$
17:        Assign label $y'$
18:      **end for**
19:    **end for**
20:    **Decode Latent Points:**
21:    **for** each class $y$ **do**
22:      $\mathbf{X}_{\text{syn}}^y = \{\mathbf{x}_1^{ly}, \mathbf{x}_2^{ly}, \ldots, \mathbf{x}_m^{ly}\}$ where $\mathbf{x}_i^{ly} = f_\theta^*(\mathbf{z}_i^{ly}), \forall \mathbf{z}_i^{ly} \in \mathbf{Z}_{\text{new}}^y$
23:    **end for**
24:    **Update Training Set:**
25:    $\mathbf{X}_{\text{aug}} \leftarrow \mathbf{X}_{\text{aug}} \cup \left(\bigcup_y \mathbf{X}_{\text{syn}}^y\right)$
26:    $\mathbf{Y}_{\text{aug}} \leftarrow \mathbf{Y}_{\text{aug}} \cup \left(\bigcup_y \{y\} \times \mathbf{X}_{\text{syn}}^y\right)$
27: **end while**

## 4. Experiments

### 4.1. Experimental setup

**Train/Test datasets:** Experiments were conducted using the UCR Time Series Archive, which comprises 120 univariate time series datasets from various applications and domains, including sensors, ECG, Motion, Spectro, etc. (a complete list of the dataset types is provided in Table 4). To guarantee an adequate amount of time series data in the datasets to train the studied models, we excluded datasets with insufficient data, retaining 102 datasets from the initial set of 120.

**Classification models:** Classifiers selected for our experiments were chosen based on the findings of (Fawaz, 2020), which reports that ResNet-50 and Fully Connected Networks (FCN) are the two most effective classifiers (out of 9 evaluated for the UCR datasets). We use the architectures from (Koonce & Koonce, 2021) and (Scabini & Bruno, 2023) for these two classifiers.

**Benchmarked DA methods:** ASCENSION is compared to six state-of-the-art DA methods, including one traditional (FAA) and five generative methods (TTS-GAN, LA, Time-DDPM, VaDE and MODALS). More details on these methods can be found in Appendix A. FAA was selected due to its comparable performance with other traditional DA methods (incl., RA and DAA), while VaDE and MODALS were chosen because of their architectural similarity to ASCENSION. TTS-GAN, Time-DDPM and LA were included as the most recent generative DA methods with publicly

available code (*cf.*, Figure 6). Benchmarking MODALS on the UCR datasets is not feasible, as its publicly available code from 2020 is no longer functional, and the authors have confirmed they do not intend to fix it. Consequently, we evaluate ASCENSION against MODALS using the HAR dataset originally used by (Cheung & Yeung, 2020).

### 4.2. Experimental Results

#### 4.2.1. PERFORMANCE EVALUATION METRICS

**Accuracy**: The ratio of correct predictions to the total number of predictions is employed as the evaluation metric. Pre- and post-augmentation classification results are gathered for each combination of the benchmarked techniques, selected classifiers, and UCR datasets. Table 1 groups the results in three categories: *(i) Augmented:* reflects the number of datasets on which the classification accuracy post-augmentation is better than pre-augmentation; *(ii) Unchanged:* refers to the datasets that do not show a significant impact ($\pm 10^{-4}\%$) of the augmentation on classifier performance, *(iii) Worsened:* aggregates the datasets where the augmentation of the train set degrades the accuracy of the classifier. Under each category we report the number of datasets and the mean classification accuracy post-augmentation for the different configurations (classifiers, DA methods). For an exhaustive list of the pre- and post-augmentation classification results, refer to Appendix B.1.

#### 4.2.2. PERFORMANCE COMPARISON ANALYSIS

Several findings can be drawn from Table 1. First, FAA demonstrates moderate mean accuracy improvements of 6.5% (ResNet) and 7.5% (FCN), but lacks consistency, with improvements observed on only 28/102 datasets (ResNet) and 13/102 datasets (FCN). Similarly, LA shows limited impact, improving accuracy on 23 datasets (ResNet) and 38 datasets (FCN), with mean improvements of 3.7% and 2.1%, respectively. On the other hand, ASCENSION achieves substantial gains, improving classification accuracy on 56/102 datasets (ResNet) and 50/102 datasets (FCN), with mean accuracy increases of 4.0% and 3.0%, respectively. Moreover, ASCENSION consistently minimizes performance deterioration, with only 30 datasets worsened for ResNet and 39 for FCN, compared to 67 and 85 datasets for FAA, respectively.

Compared to Time-DDPM and VaDE, ASCENSION achieves a balanced trade-off between maximizing the number of datasets improved and minimizing those with worsened performance. Time-DDPM, while achieving the highest mean accuracy improvement (17.8% for ResNet and 15.8% for FCN), suffers from significant performance deterioration on 62/102 datasets (ResNet) and 58/102 datasets (FCN), indicating overfitting to a subset of datasets. In contrast, ASCENSION's consistent performance across both

Table 1: Results of our empirical benchmark study on the 102 UCR datasets. The table summarizes the number of datasets with improvement (Augmented), no change (Unchanged), and deterioration (Worsened) in classification accuracy for each DA method. The mean accuracy change ($\overline{\text{Acc}}$) is provided for each category. An upward arrow (↑) indicates that higher values are preferable, while a downward arrow (↓) signifies that lower values are better. Bold values denote the **best performance**, and underlined values indicate the _second best. ASCENSION improves the classification accuracy for the highest number of datasets and produces the fewest cases of performance reduction, demonstrating its effectiveness in enhancing classification accuracy across the datasets.

| | DA method | Augmented | | Unchanged | | Worsened | | ↑Total | |
|---|---|---|---|---|---|---|---|---|---|
| | | ↑Nb$_{datasets}$ | ↑$\overline{\text{Acc}}$ | Nb$_{datasets}$ | $\overline{\text{Acc}}$ | ↓Nb$_{datasets}$ | ↑$\overline{\text{Acc}}$ | Nb$_{datasets}$ | ↑$\overline{\text{Acc}}$ |
| ResNet | FAA | 28 | 6.5% | 7 | **0%** | 67 | -9.1% | 102 | -4.2% |
| | LA | 23 | 3.7% | 12 | **0%** | 67 | -3.3% | 102 | -1.3% |
| | TTS-GAN | 41 | 2.2% | 10 | **0%** | 51 | -8.9% | 102 | -3.6% |
| | Time-DDPM | 38 | **17.8%** | 2 | **0%** | 62 | -22.2% | 102 | -6.8% |
| | VaDE | **57** | 3.1% | 8 | **0%** | 37 | -7.7% | 102 | -1.1% |
| | **ASCENSION** | 56 | 4.0% | 16 | **0%** | **30** | **-1.7%** | 102 | **1.7%** |
| FCN | FAA | 13 | 7.5% | 4 | **0%** | 85 | -15.8% | 102 | -12.2% |
| | LA | 38 | 2.1% | 18 | **0%** | 46 | -2.3% | 102 | -0.3% |
| | TTS-GAN | 31 | 2.2% | 13 | **0%** | 58 | -7.5% | 102 | -3.6% |
| | Time-DDPM | 43 | **15.8%** | 1 | **0%** | 58 | -24.0% | 102 | -7.0% |
| | VaDE | 35 | 2.8% | 16 | **0%** | 51 | -6.7% | 102 | -2.4% |
| | **ASCENSION** | **50** | 3.0% | 13 | **0%** | **39** | **-1.4%** | 102 | **1.0%** |

Table 2: Acc. comparison on HAR dataset used by (Cheung & Yeung, 2020) to assess MODALS

| Method | Accuracy (%) |
|---|---|
| ASCENSION$_{ResNet-Emb}$ | **93.42** |
| MODALS | 91.87 |
| No Augmentation | 88.64 |

ResNet and FCN backbones demonstrates its scalability and versatility for enhancing classification tasks.

In Table 2, we compare to MODALS on the HAR dataset, ASCENSION further enhances performance. While MODALS improves the baseline classification (without augmentation) by 3.23%, ASCENSION increases this improvement by +4.78%, further advancing accuracy beyond the baseline.

### 4.2.3. EMBEDDED CLASSIFIER PERFORMANCE

The ASCENSION framework supports various classifier architectures due to its modularity. Leveraging this flexibility, we also assess ASCENSION's performance with a modified classifier setup. In Table 3, we present the evaluation results for: (i) ASCENSION's standard embedded classifier, denoted as ASCENSIONEmbCl., and (ii) a hybrid approach combining ASCENSION's embedded classifier with the studied classifiers, referred to as ASCENSION$c$-EmbCl., where $c \in$ ResNet, FCN in our experiments. The augmentation effect is quantified as the difference between: (i) The highest baseline accuracy achieved by either the VAE's

classifier or the standalone classifier $c$, and (ii) the highest accuracy recorded for ASCENSION$_{EmbCl.}$ or classifier $c$, computed as follows:

$$\text{Acc}_{\text{ASCENSION}_{c\text{-EmbCl.}}} = \max(\text{Acc}_{\text{ASCENSION}_{\text{EmbCl.}}}, \text{Acc}_c) - \max(\text{Acc}_{\text{Baseline}}, \text{Acc}_{\text{VAE}}) \quad (4)$$

Table 3 shows that ASCENSION$_{ResNet-Emb}$ achieves the highest accuracy gain (3.7% on 76 datasets) but also has the largest accuracy drop (-5.7% on 14 datasets). ASCENSION$_{EmbCl.}$ offers a more stable performance (1.9% improvement) with minimal degradation (-1.6%). ASCENSION$_{FCN-Emb}$ provides moderate gains (2.9%) with a balanced trade-off. Overall, a more complex architecture such as ResNet is likely to maximize improvement but introduces variability, while FCN and the standard classifier ensure more stable performance.

### 4.2.4. HYPERPARAMETERS SENSITIVITY ANALYSIS

A key feature of ASCENSION is its controllable progressive expansion mechanism for exploring the latent space. Adjusting the scaling factor parameter $\alpha$ – *which influences how distributions are flattened, see section 3.1* – and determining the number of iterations are essential for optimizing the method's effectiveness. These two parameters must be carefully balanced to maintain sufficient separation between distributions while allowing for adequate exploration.

**Analysis methodology:** We conducted a study that varies $\alpha$ (from 1 to 5) and the number of iterations (from 1 to 9) to

Table 3: The table summarizes the number of datasets with improvement (Augmented), no change (Unchanged), and deterioration (Worsened) in classification accuracy for each inherent classifier architecture. The mean accuracy change ($\overline{\text{Acc}}$) is provided for each category. An upward arrow (↑) indicates that higher values are preferable, while a downward arrow (↓) signifies that lower values are better. Bold values denote the **best performance**.

| Embedded Classifier | Augmented | | Unchanged | | Worsened | | ↑Total | |
|---|---|---|---|---|---|---|---|---|
| | ↑$\text{Nb}_{\text{datasets}}$ | ↑$\overline{\text{Acc}}$ | $\text{Nb}_{\text{datasets}}$ | $\overline{\text{Acc}}$ | ↓$\text{Nb}_{\text{datasets}}$ | ↑$\overline{\text{Acc}}$ | $\text{Nb}_{\text{datasets}}$ | ↑$\overline{\text{Acc}}$ |
| ASCENSION$_{\text{Emb.}}$ | 65 | 1.9% | 24 | **0%** | 24 | ***-1.6%*** | 102 | 0.8% |
| ASCENSION$_{\text{ResNet-Emb}}$ | **76** | **3.7%** | 12 | **0%** | **14** | -5.7% | 102 | **2.1%** |
| ASCENSION$_{\text{FCN-Emb.}}$ | 60 | 2.9% | 28 | **0%** | 14 | -1.7% | 102 | 1.2% |

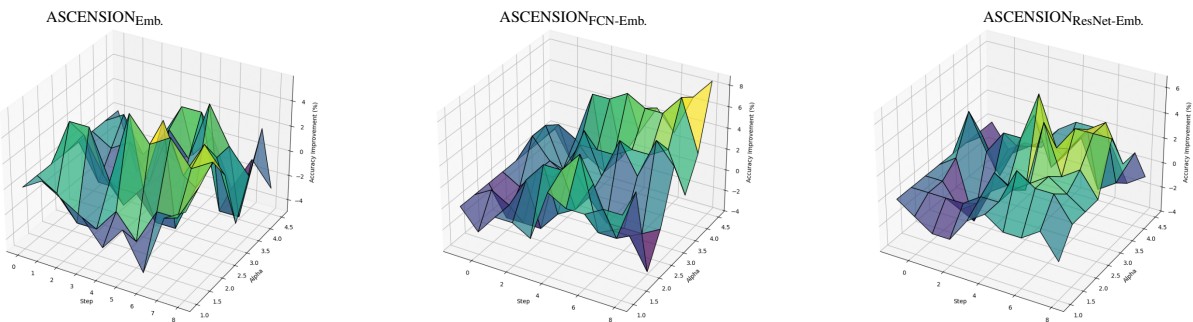

Figure 2: Analysis of accuracy augmentation as a function of the parameter $\alpha$ and the number of augmentation steps for the **Ham** dataset. The results suggest that clearly defining optimal values for $\alpha$ and the maximum number of iterations is challenging. However, it is evident that $\alpha$ should remain above 1, and a minimum threshold of approximately 3 iterations is deemed acceptable. A comprehensive grid search may be warranted to identify the optimal parameter values. More examples can be found in appendix C.

assess their impact on accuracy improvement and determine whether convergence occurs.

**Results:** Figure 2 presents the results for ASCENSION$_{\text{EmbCl.}}$, ASCENSION$_{\text{ResNet-EmbCl.}}$, and ASCENSION$_{\text{FCN-EmbCl.}}$ using the **Ham** dataset from the UCR archive (additional examples can be found in Appendix C). The augmentation process remains relatively stable even with high $\alpha$ values, supporting our hypothesis that the distribution borders reduce the sensitivity of ASCENSION to changes in $\alpha$. Appendix C offers similar analyses across various UCR datasets, showing that increasing $\alpha$ can enhance boundary exploration but may reduce performance if $\alpha$ is too large. Based on our experiments, selecting $\alpha$ in the range $[1, 3]$ provides a good balance.

### 4.2.5. OPERATIONAL EFFICIENCY ANALYSIS

Section 4.2 has empirically evidenced that ASCENSION generally outperforms traditional and generative state-of-the-art DA methods for TSC across most datasets. However, a substantial proportion of datasets (30% to 50%) do not exhibit improved classification performance, and in some cases, performance even deteriorates (see the Unchanged and Worsened columns in Table 1). A comprehensive list of these datasets can be found in Appendix B.1. To address

this, we propose an analysis to determine which types of data – *characterized by their specific features* – benefit the most from augmentation and which require minimal or no augmentation.

**Feature extraction:** We use the CATCH22 time series feature set introduced by (Lubba et al., 2019) to characterize the datasets (comprising 22 features in total), adding the ratio of train/test split and the distribution discrepancy ratio between train and test (cf., Appendix E.1). A description of these 24 features (F1-F24) is provided in Appendix F.

**Analysis methodology:** By averaging the features of the time series in each dataset, we identify the datasets that are most and least amenable to benefit from augmentation. Subsequently, we analyze the impact of augmentation on the classification performance of these datasets to determine the most influential features. To measure feature importance, we employ a random forest model with a high number of estimators with low depth to the mean of F1-F24 to predict augmentation for the benchmarked DA methods.

**Results:** Figure 3 shows that each method is strongly tied to specific features such as FAA to F10 (degree of periodic patterns within the dataset), TTS-GAN to F7 which is related to rapid fluctuation in the time series. Moreover, features F23 and F24 (respectively representing the train/test ratio

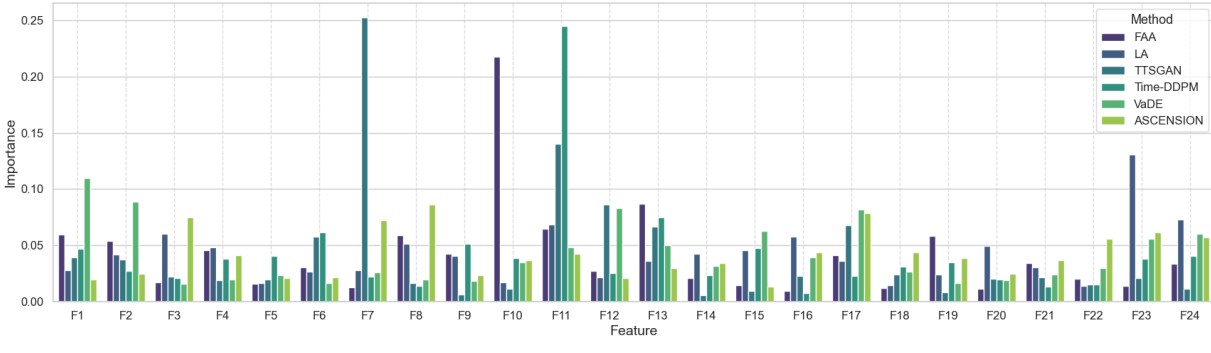

Figure 3: Feature importance derived from a random forest model applied to the 24 features (F1-F24, cf. Appendix F.) F7 indicates rapid fluctuations in the time series, F10 indicates the repetition of pattern in the time series, F11 estimates the differences in distances between successive points in a 2-dimensional embedding space, F23 is the ratio of train and test data in the dataset and, F24 is the discrepancy in distance between the train and test set distributions, see Appendix E.1.

of data and discrepancy in distance between the Train and Test set distributions, *cf.* Appendix E.1) are tied to methods such as LA and ASCENSION.

To analyze the impact of increasing train-test discrepancy ratios on classification performance, Figure 4 presents the cumulative performance improvement (%) as a function of F24 (see Appendix E.1). The 102 UCR datasets are arranged in ascending order of discrepancy. While other DA methods experience performance degradation as discrepancy increases, ASCENSION sustains positive performance and even exhibits a slight improvement.

### 4.2.6. QUALITATIVE STUDY ON THE RISK OF EXTRAPOLATION

To qualitatively assess our extrapolation process, we introduce a *class assignment confidence* measure for each generated latent sample set $Z$. Specifically, we sample a class $y$ from $\{1, 2, \ldots, Y\}$ and define its confidence as:

$$\mathbf{P}_y\left(\mathcal{L}(y|Z) = \max_{k \text{ in } \{1,2,\ldots,Y\}} (\mathcal{L}(k|Z))\right), \quad (5)$$

where $\mathcal{L}(y \mid Z)$ denotes the likelihood that $Z$ belongs to the distribution associated with class $y$. We empirically compute this probability by sampling $n = 1000$ points and measuring the proportion of samples most likely to originate from the intended class.

It is worth noting that ASCENSION applies the same likelihood-based filtering criterion before incorporating generated samples into the final training set. Therefore, this confidence metric indicates how often a sample aligns with its target class before any filtering removes unreliable points. As a result, our measure serves as a valuable yet inherently qualitative indicator of the model's initial ability to generate class-consistent samples.

As shown in Figure 5, contrary to initial expectations, class assignment confidence does not significantly decline throughout the expansion process. This indicates that confidence retention is more influenced by the intrinsic characteristics of each dataset rather than the expansion itself. For a more detailed analysis, readers can refer to Appendix D.

## 5. Conclusion & Future Works

This paper introduced ASCENSION, a novel VAE-based DA method for TSC that integrates a controllable and progressive class boundary expansion mechanism. Unlike existing generative DA methods, which primarily rely on interpolating within the existing training distribution, ASCENSION enables controlled extrapolation, preserving intra-class coherence and enabling the user to monitor inter-class separation. By leveraging a probabilistic latent space structure, ASCENSION effectively generates synthetic samples that enhance classification performance across a broad range of time series datasets.

Our benchmarking analysis on 102 UCR datasets highlights ASCENSION's ability to deliver consistent performance improvements. Compared to six state-of-the-art DA methods—FAA, LA, TTS-GAN, Time-DDPM, VaDE, and MODALS—ASCENSION achieved the highest overall classification gains, improving accuracy in 55% of datasets with ResNet and 49% with FCN, while limiting performance degradation to only 29% and 38%, respectively. Additionally, our analysis of DA effectiveness factors reveals that ASCENSION performs particularly well in scenarios where the discrepancy between training and test data is relatively high, whereas other methods experience a sharp decline in effectiveness under such conditions. This finding is particularly significant, as real-world applications often involve variations in train-test distribution discrepancies (see e.g.

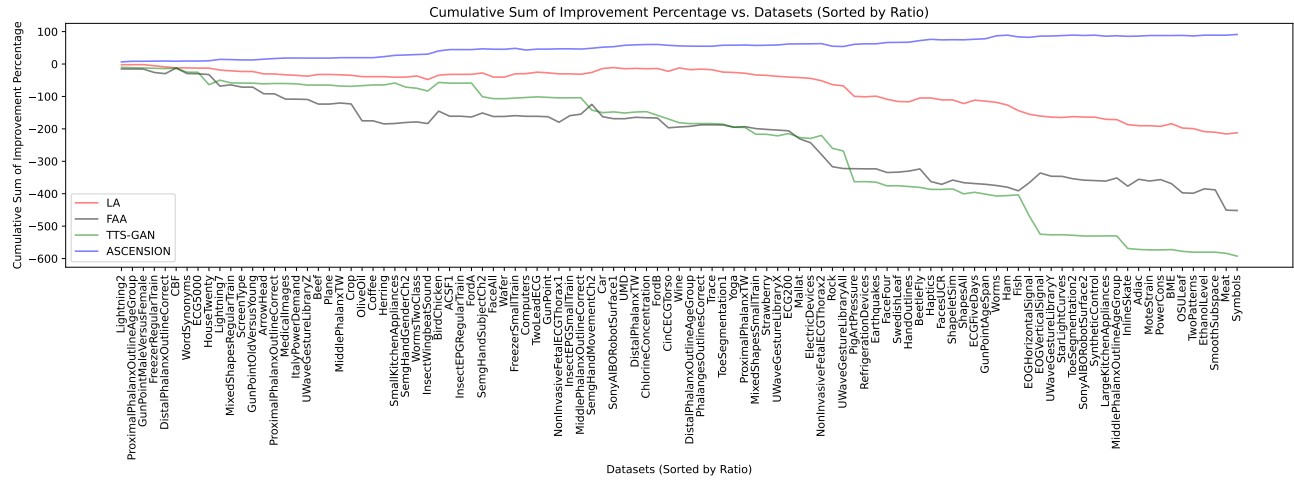

Figure 4: Cumulative performance improvement (%) as a function of F24, which represents the train-test discrepancy ratio (see Appendix E.1). The 102 UCR datasets are ordered in increasing discrepancy. While other data augmentation (DA) methods show performance degradation as discrepancy rises, ASCENSION maintains a positive performance trend and even demonstrates a slight improvement.

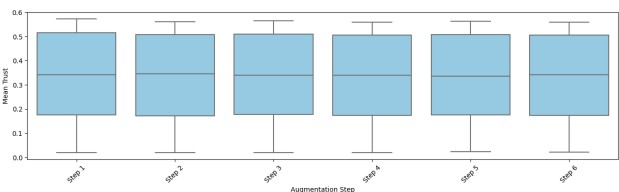

Figure 5: Class confidence distribution over the different augmentation steps. Class assignment confidence does not significantly decline throughout the expansion process.

(Koh et al., 2021)), making ASCENSION a valuable asset for practical deployment.

**Limitations & Future work:** While ASCENSION advances generative DA for time series, certain limitations remain. The latent space expansion mechanism requires careful tuning of parameters such as the scaling factor, the number of augmentation steps, and the step size. Automating these hyperparameter selections based solely on training data could be a promising direction for future work. Although ASCENSION ensures class-consistent sampling, incorporating domain-specific priors could further refine boundary expansions. Additionally, ASCENSION's framework could be extended to other types of sequential data (e.g., natural language, spatio-temporal data) as well as non-sequential domains (e.g., images). Exploring alternative clustering methods, sampling strategies, and expansion mechanisms *beyond a single $\alpha$ factor* – could further improve its adaptability and effectiveness across diverse applications.

## 6. Software and Data

The UCR time series archive can be found at https://www.cs.ucr.edu/~7Eeamonn/time_series_data_2018/. We detailed exact implementation details and provide code to produce our results on an anonymous github page at https://github.com/ASCENSION-PAPER

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

## A. Related work

(Iglesias et al., 2023b) and (Iwana & Uchida, 2021) divide DA for time series into two categories: Traditional *vs.* Generative DA methods. Figure 6 offers an overview of the evolution of these methods.

**Traditional DA methods**, such as window slicing, jittering, and scaling (Iglesias et al., 2023a), are primarily adapted from computer vision and rely on transformation strategies like cropping, rotation, scaling, drifting, and so forth. However, the complex nature of time series data often renders these methods sub-optimal, as they can disrupt the semantic integrity of the original data. For instance, while a slightly flipped image of a cat remains recognizable, reversing the time axis of an electrocardiogram sequence can render it meaningless. In response to these challenges, more advanced DA techniques were developed to automate the sequence of transformations to be performed. A first method, named **AutoAugment (AA)** (Cubuk et al., 2019), uses reinforcement learning to explore transformation pipelines/policies. A second method named **Fast AutoAugment (FAA)** (Lim et al., 2019) uses density matching for a faster search strategy, eliminating the need for backpropagation. Subsequent methods such as **RandAugment** (Cubuk et al., 2020), **Deep AutoAugment** (Zheng et al., 2022), and **Trivial Augment** (Müller & Hutter, 2021) were introduced to further simplify and refine the augmentation search strategy. RandAugment streamlines the augmentation process by removing the exhaustive search phase, instead applying a fixed number of random transformations

with adjustable magnitudes. Deep AutoAugment incorporates a deep reinforcement learning model that dynamically combines transformation policies based on the specific characteristics of the dataset. Trivial Augment introduces an even simpler approach by applying a minimal set of random transformations, emphasizing ease of use and computational efficiency. Despite all these advancements, all these methods rely on predefined transformations, which is suboptimal for preserving intra-class consistency and the semantic characteristics of the original time series data, thereby limiting the effectiveness of data augmentation.

**Generative DA models** such as Generative Adversarial Networks (GANs) (Goodfellow et al., 2020), diffusion models (Yang et al., 2023a), and VAEs (Kingma & Welling, 2013) represent powerful techniques capable of learning a probabilistic representation of data distributions. These models can generate time series data that retain the temporal dependencies, semantic consistency, and class-specific characteristics of the original datasets (Fu et al., 2020). For example, using a representation layer, as introduced by (Liu et al., 2022), provides an abstraction that is crucial when dealing with time series data. **TimeGAN** (Zhang et al., 2022) has been specifically designed for time series, which has shown significant improvements in generating high-quality synthetic sequences and augmenting low-quality datasets. Likewise, **TS-GAN** (Yang et al., 2023b) develop a LSTM-based GAN architecture with an sequential-squeeze-and-excitation to better capture time-dependence between the current and past moments in each dimensions. TS-GAN is particulary proposed to generate augmented sensor-based health data to improve Deep Learning (DL) classification models and evaluated on 3 health time series datasets. **TTS-GAN** (Li et al., 2022) adapt the traditional GAN architecture using a transfomer-encoder architecture that can deal with long range dependencies in time sequences. It shows strong performance in generating realistic data across three datasets: a simulated dataset, a human acuity recognition dataset, and an ECG dataset. However, GANs training process is very unstable and is very senstive to hyperparameters. It also suffers from issue as mode collapse that can limit the variety of generated samples and can possibly generate unrealistic data (Lei et al., 2019). **LatentAugment** (Tronchin et al., 2023) learns a low-level representation of initial data, noising around learned points and then decoding them to produce newly generated and semantically close data. More recently, (Seon et al., 2024) proposed **LISGAN**, a GAN-based architecture to augment time series data in the context of class imbalance by adjusting the loss with mutual information term and using a spectral normalization. LISGAN generates high quality synthetic data and significantly increases classification performance with industrial internet of things datasets. Diffusion models, a more recent class of generative models, have garnered significant attention for

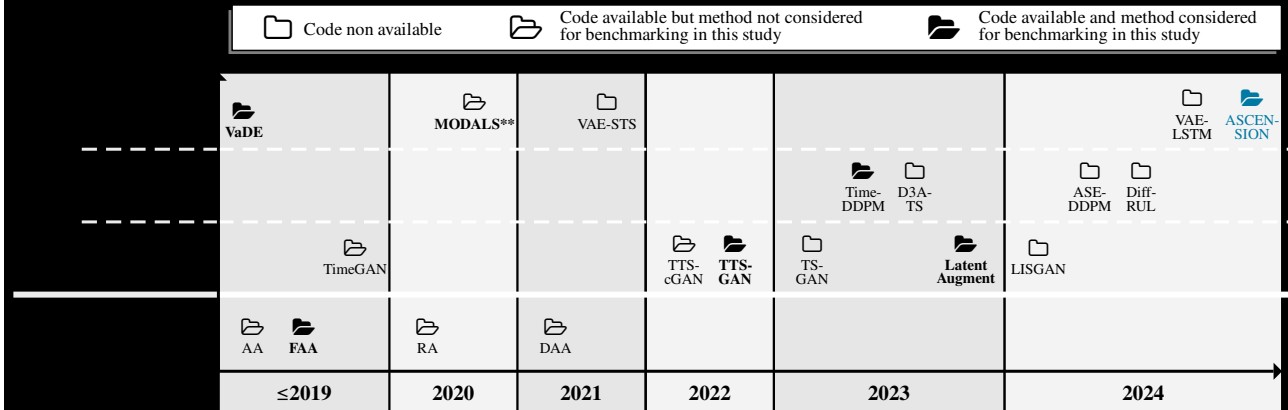

Figure 6: Overview of the evolution of state-of-the-art data augmentation methods for time series (traditional vs. generative). **MODALS: Although code was made available (4 years ago), it is currently non-functional; we have contacted the authors of MODALS (Cheung & Yeung, 2020) for the source code, but they informed us that it is no longer operational and cannot be repaired without substantial re-coding.

their capability to model complex data distributions. Unlike GANs, which rely on adversarial training, diffusion models generate data by progressively refining noise toward the target data distribution. This denoising approach has yielded remarkable results in high-fidelity image generation, as seen with models like DALL·E 2, Imagen, and Flux. Recently, starting in 2023, several diffusion model-based DA methods for time series have emerged, including **ASE-DDPM** (Liu et al., 2024) for addressing imbalanced time series classification, **DiffRUL** (Wang et al., 2024) for enhancing remaining useful life predictions, **D3A-TS** (Solis-Martin et al., 2023) aimed at improving synthetic sample quality through meta-attribute conditioning, and **Time-DDPM**, which integrates a diffusion denoising probabilistic model with CNN-LSTM networks to enhance sample quality. While diffusion models provide stable outputs, they face challenges with long-range predictions, error accumulation, and slow inference (Feng et al., 2024), which can limit their practical applications. VAEs offer several advantages over GANs and diffusion models. Their probabilistic nature allows for explicit control over the diversity and quality of generated samples through manipulation of the latent space, as evidenced in (Cheung & Yeung, 2020). This helps preserve the intra-class consistency and semantic characteristics of the original data. Additionally, VAEs are less prone to collapse compared to GANs and are less computationally expensive than both GANs and diffusion models (Thanh-Tung & Tran, 2020). To our knowledge, the first VAE-based generative DA model relying of clustering, named **VaDE**, was introduced in (Jiang et al., 2016). The authors integrate a prior GMM fitting to the VAE training, enabling realistic samples generation for any specified cluster, without using supervised information during training. **MODALS**, was introduced by (Cheung & Yeung, 2020) and represents the closest architectural approach to ASCENSION. It was the first study to investigate

the expansion of class boundaries during synthetic data generation, although it does not offer a method for controlling this expansion. Recently, (Dang et al., 2024) introduced **VAE-LSTM**, which is used to augment an inertial sensor dataset due to limited data availability, with the goal of enhancing classification performance. However, this approach does not explore the expansion of class representations in the latent space, as proposed in ASCENSION.

# B. Enlarged experimental result analysis

## B.1. Enlarged classification performance

This section offers a more comprehensive analysis of the results. The 102 datasets from the UCR time series classification repository are grouped into 11 distinct categories (domains/applications), as summarized in Table 4.

A detailed breakdown of our experimental results is presented in Table 5 and Table 6.

# C. Enlarged hyperparameters sensitivity analysis

Figures 7 to 16 show 3D plots of classifier performance as a function of $\alpha$ and the number of iterations for ASCENSION$_{EmbCl}$, FCN, and ResNet, across representative datasets from each category of the UCR archive. The name of each category and their representative datasets are detailed in Table 4.

$\alpha$ **parameter:** As discussed in section 4.2.4, performance improvement relation to $\alpha$ seems difficult to generalize while remaining relatively stable. Increasing $\alpha$ can lead to better boundary exploration, as shown in Figures 11 and 10 but can also make the performance drop for too high

Table 4: UCR dataset types along with the selected representative datasets

| Type | Representative dataset | Description |
|------|------------------------|-------------|
| Device | ACSF1 | Measurements of alternating current signals for predictive maintenance |
| ECG | ECG200 | Electrocardiogram (ECG) readings used to detect heart abnormalities |
| EOG | EOGVerticalSignal | Electrooculography (EOG) signals capturing eye movement patterns |
| EPG | InsectEPGRegularTrain | Electrical penetration graph (EPG) signals capturing insect feeding behavior |
| Image | BeetleFly | Shape-based image classification of beetle and fly outlines |
| Motion | Worms | Motion sensor data capturing worm movements for classification |
| Power | PowerCons | Power consumption measurements for energy usage |
| Sensor | Car | Sensor readings collected from a car, used for detecting driving conditions |
| Simulated | UMD | Simulated control processes data |
| Spectro | Ham | Spectroscopy data to identify types of ham based on chemical properties |
| Spectrum | SemgHandMovementCh2 | Electromyography (EMG) data of hand movements, recorded across channels |

Table 5: Mean Improvement per Dataset Type

| Type | FAA | | LA | | Time-DDPM | | TTS-GAN | | VaDE | | ASCENSION | |
|------|-----|-----|-----|-----|-----|-----|-----|-----|-----|-----|-----|-----|
| | $\uparrow$Nb$_{Datasets}$ | $\uparrow\overline{Acc}$ | $\uparrow$Nb$_{Datasets}$ | $\uparrow\overline{Acc}$ | $\uparrow$Nb$_{Datasets}$ | $\uparrow\overline{Acc}$ | $\uparrow$Nb$_{Datasets}$ | $\uparrow\overline{Acc}$ | $\uparrow$Nb$_{Datasets}$ | $\uparrow\overline{Acc}$ | $\uparrow$Nb$_{Datasets}$ | $\uparrow\overline{Acc}$ |
| Device | 1/8 | 7.7% | 2/8 | 3.1% | 4/8 | **20.3%** | 3/8 | 1.1% | 3/8 | 0.7% | **7/8** | 2.2% |
| ECG | 1/6 | **14.2%** | **3/6** | 0.2% | 2/6 | 3.8% | **3/6** | 1.6% | 2/6 | 0.3% | 2/6 | 0.1% |
| EOG | 0/2 | 0.0% | **1/2** | 2.8% | **1/2** | **35.8%** | 0/2 | 0.0% | 0/2 | 0.0% | 0/2 | 0.0% |
| EPG | 0/2 | 0.0% | 0/2 | 0.0% | 0/2 | 0.0% | 0/2 | 0.0% | 0/2 | 0.0% | 0/2 | 0.0% |
| Image | 2/30 | 13.2% | 10/30 | 2.2% | 13/30 | **16.4%** | 10/30 | 3.2% | 11/30 | 4.3% | **14/30** | 1.8% |
| Motion | 2/14 | 2.8% | **10/20** | 1.3% | 9/20 | **13.2%** | 1/20 | 0.8% | 9/20 | 1.5% | 8/20 | 1.0% |
| Power | 0/1 | 0.0% | **1/1** | **3.9%** | 0/1 | 0.0% | **1/1** | 2.8% | **1/1** | 1.7% | **1/1** | 2.2% |
| Sensor | 2/19 | 5.4% | **7/19** | 2.0% | 5/19 | **17.2%** | 6/19 | 2.4% | 6/19 | 3.2% | **7/19** | 1.2% |
| Simulated | 3/8 | 3.5% | 2/8 | 5.3% | 1/8 | **12.6%** | **5/8** | 1.0% | 0/8 | 0.0% | 2/8 | 5.7% |
| Spectro | 2/8 | **11.3%** | 1/8 | 0.4% | 4/8 | 6.1% | 2/8 | 2.9% | 1/8 | 1.7% | **7/8** | 9.9% |
| Spectrum | 0/4 | 0.0% | 1/4 | 6.0% | **4/4** | **24.7%** | 0/4 | 0.0% | 2/4 | 7.1% | 2/4 | 5.3% |

Table 6: Mean Negative Impact per Dataset Type

| Type | FAA | | LA | | Time-DDPM | | TTS-GAN | | VaDE | | ASCENSION | |
|------|-----|-----|-----|-----|-----|-----|-----|-----|-----|-----|-----|-----|
| | $\downarrow$Nb$_{Datasets}$ | $\uparrow\overline{Acc}$ | $\downarrow$Nb$_{Datasets}$ | $\uparrow\overline{Acc}$ | $\downarrow$Nb$_{Datasets}$ | $\uparrow\overline{Acc}$ | $\downarrow$Nb$_{Datasets}$ | $\uparrow\overline{Acc}$ | $\downarrow$Nb$_{Datasets}$ | $\uparrow\overline{Acc}$ | $\downarrow$Nb$_{Datasets}$ | $\uparrow\overline{Acc}$ |
| Device | 7/8 | −8.8% | 5/8 | **−2.5%** | 4/8 | −23.6% | 5/8 | −9.1% | 5/8 | −3.0% | **1/8** | −4.0% |
| ECG | 5/6 | −27.9% | 3/6 | −3.7% | 4/6 | −16.9% | 3/6 | **−1.9%** | 3/6 | 18.3% | **2/6** | −4.3% |
| EOG | 2/2 | −21.1% | **1/2** | −6.6% | **1/2** | −17.9% | 2/2 | −32.2% | 2/2 | −11.3% | 2/2 | **−1.2%** |
| EPG | **0/2** | 0.0% | **0/2** | 0.0% | 2/2 | −11.1% | **0/2** | 0.0% | **0/2** | 0.0% | **0/2** | 0.0% |
| Image | 27/30 | −17.6% | 15/30 | −1.9% | 17/30 | −20.7% | 17/30 | −5.9% | **13/30** | −11.1% | 14/30 | **−1.3%** |
| Motion | 11/14 | −13.6% | **2/14** | −1.4% | 5/14 | −27.8% | 11/14 | −11.1% | 4/14 | −2.5% | 5/14 | −2.0% |
| Power | 1/1 | −2.8% | **0/1** | **0.0%** | 1/1 | −88.5% | **0/1** | **0.0%** | **0/1** | **0.0%** | **0/1** | **0.0%** |
| Sensor | 17/19 | −11.1% | 10/19 | −1.4% | 14/19 | −28.7% | 11/19 | −3.1% | 11/19 | −2.5% | **6/19** | −0.4% |
| Simulated | 5/8 | −15.9% | 3/8 | −1.6% | 6/8 | −18.3% | **1/8** | −1.4% | 6/8 | −8.5% | 5/8 | **−0.8%** |
| Spectro | 6/8 | −32.4% | 4/8 | −5.1% | 4/8 | −25.2% | 4/8 | −3.2% | 5/8 | −2.0% | **1/8** | **−0.5%** |
| Spectrum | 4/4 | −3.4% | 3/4 | −2.0% | **0/4** | **0.0%** | 4/4 | −12.2% | 2/4 | −3.9% | 2/4 | −0.6% |

values of $\alpha$. While pinpointing the exact $\alpha$ values and iterations for optimal results across all datasets is not trivial, the general trend suggests selecting $\alpha \in [1, 3]$ to expand class boundaries without venturing into areas that risk class overlap, which could negatively impact classification accuracy.

**Number of iterations:** In Figures 10-12, and 14, we observe that a higher number of iterations can have either a positive or negative impact on performance, whereas in Figure 7, the number of iterations does not play a significant role in performance improvement. This ambivalent behavior is closely related to the class distribution within the dataset. As the number of iterations increases, classes in the latent space may become closer due to the increase in the $\alpha$ pa-

rameter at each iteration, which leads to the expansion of covariances $\alpha\Sigma_k$ (*cf.*, Figure 1). Therefore, we recommend carefully adjusting the number of iterations in relation to the chosen $\alpha$ parameter.

ASCENSION$_{Emb.}$     ASCENSION$_{FCN-Emb.}$     ASCENSION$_{ResNet-Emb.}$

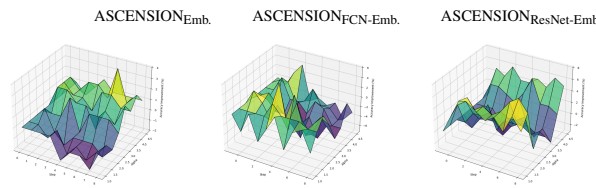

Figure 7: **ECG:** Classifier performance against $\alpha$ and iteration number for **ECG200** dataset.

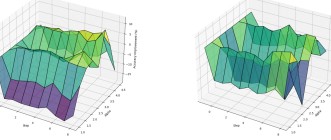

Figure 8: **EOG:** Classifier performance against $\alpha$ and iteration number for **EOGVerticalSignal**.

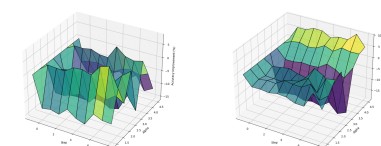

Figure 9: **Hemodynamics:** Classifier performance against $\alpha$ and iterations for **PigArtPressure**.

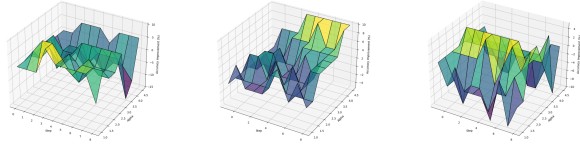

Figure 10: **Image:** Classifier performance against $\alpha$ and iteration number for **BeetleFly** dataset.

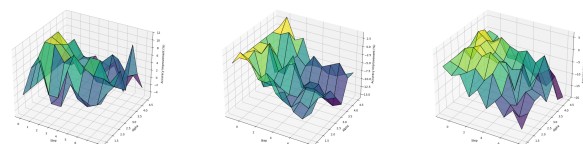

Figure 11: **Motion:** Classifier performance against $\alpha$ and iteration number for **Worms** dataset.

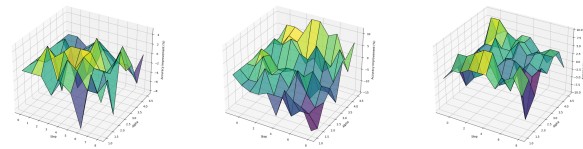

Figure 12: **Sensor:** Classifier performance against $\alpha$ and iteration number for **Car** dataset.

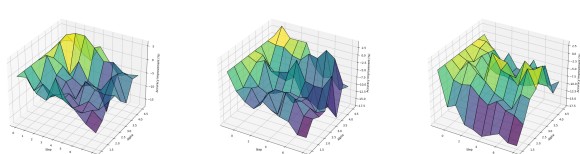

Figure 13: **Simulated:** Classifier performance against $\alpha$ and iteration number for **UMD** dataset.

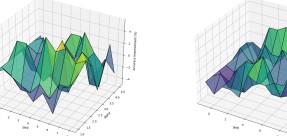

Figure 14: **Spectro:** Classifier performance against $\alpha$ and iteration number for **Ham** dataset.

$\text{ASCENSION}_{\text{Emb.}}$  $\text{ASCENSION}_{\text{FCN-Emb.}}$  $\text{ASCENSION}_{\text{ResNet-Emb.}}$

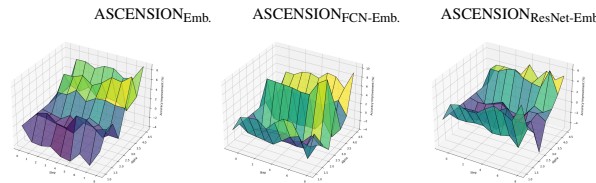

Figure 15: **Spectrum:** Classifier performance against $\alpha$ and iteration number for **SemgHandMovementCh2** dataset.

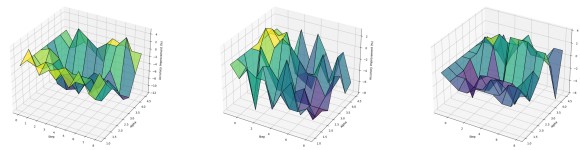

Figure 16: **Device:** Classifier performance against $\alpha$ and iteration number for **ACSF1** dataset.

## D. Enlarged analysis of the class assignment confidence

All following figures of this section have been computed after removing outlier data samples.

Both Figures 17 and 18 show a complex relationship between confidence and performance. A slight positive correlation appears to be present, however it is clear that no linear or polynomial relationship exists between the two.

From the previous analysis, we perform a clustering using DBSCAN to extract patterns. Figures 19 and 20 reveal two main clusters. As mentioned previously, we infer that these clusters may depend on the initial conditions of the augmentation, that is to say, the dataset and its characteristics.

We validate this hypothesis by computing the feature importances of the dataset's features defined in Appendix F in regards to predicting confidence through a Random Forest Regressor built with a high number of shallow trees. The negative or positive characteristic of the importance is then computed using a correlation matrix.

The results in Figure 21 show five features with predominant importance. The contrast in these importance allows us to

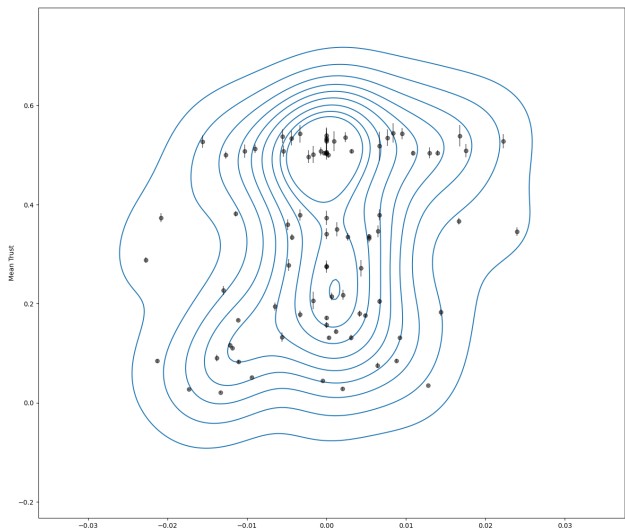

Figure 17: **Trust in regards to performance for FCN :** Overview of the relationship between mean trust over the augmentation steps and final performance.

.

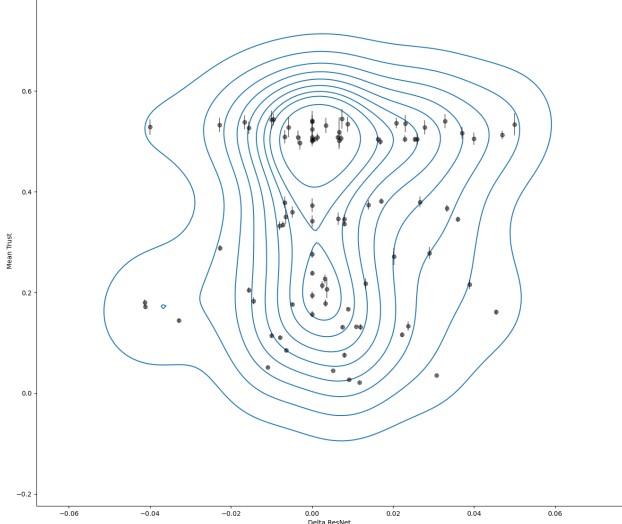

Figure 18: **Confidence in regards to performance for ResNet :** Overview of the relationship between mean confidence over the augmentation steps and final performance.

.

validate the hypothesis that some datasets features seem to have a relationship with the confidence of the expansion mechanism.

(The full tables of results are available in the supplementary materials in csv and json format.)

## E. Performance metric formalization

### E.1. Discrepancy in distance between training and test sets

E.1.1. FORMALIZATION

To estimate the discrepancy in distance between the training and test sets, we compute the mean intra-class distance across all classes using DTW as the distance metric. Let $\mathcal{X}_k = x_{k,1}, x_{k,2}, \ldots, x_{k,n_k}$ represent the set of generated samples belonging to class $k$, and $d_k$ be the mean intra-class distance for class $k$, defined as:

$$d_k = \frac{1}{n_k} \sum_{i=1}^{n_k} \mathrm{DTW}(x_{k,i}, \mu_k) \qquad (6)$$

where $\mu_k$ is the mean of the samples in class $k$ (computed using DTW barycenter averaging, where applicable). The overall dispersion $D$ of the dataset is then defined as the mean intra-class variance across all $K$ classes:

$$D = \frac{1}{K} \sum_{k=1}^{K} d_k \qquad (7)$$

To estimate the discrepancy between the training and test

datasets, we compute the ratio between the dispersion of the test set $D_{\text{test}}$ and the diversity of the train set $D_{\text{train}}$. This ratio $V$ is defined as:

$$V = \frac{D_{\text{test}}}{D_{\text{train}}} \qquad (8)$$

The discrepancies ratio $V \approx 1$ indicates similar diversity between the train and test sets, while deviations from 1 suggest more diversity in the training set ($V < 1$) or in the test set ($V > 1$).

A dataset where the ratio $V > 1$ is considered to be more challenging for usual generative techniques, as the train set does not accurately represent the test set in these cases. As such the datasets at the far right in

E.1.2. EXPERIMENTAL RESULTS

The discrepancy ratio of the 102 UCR datasets have been plotted in an ascending order in Figure 22. Le us consider three datasets with extreme ratios: **(i) Discrepancy toward test:** Dataset `Car` (1.51); **(ii) No discrepancy:** Dataset `ECGFiveDays` (1.01); **(iii) Discrepancy toward train:** Dataset `EOGVerticalSignal` (0.77).

Detailed results of the discrepancies across datasets are available in Table 7

## F. Time series features

In this section, we describe the 22 time series features (Catch22) presented in (Lubba et al., 2019), and the two

Table 7: Discrepancy Metrics Across Datasets

| $Dataset$ | Ratio | $Dispersion_{TEST}$ | $Dispersion_{TRAIN}$ |
|---|---|---|---|
| HandOutlines | 0.46 | $1.50 \times 10^2$ | $1.39 \times 10^2$ |
| GesturePebbleZ2 | 0.66 | $3.09 \times 10^1$ | $3.02 \times 10^1$ |
| ShakeGestureWiimoteZ | 0.71 | $5.36 \times 10^2$ | $6.04 \times 10^2$ |
| GestureMidAirD1 | 0.75 | $4.18 \times 10^2$ | $4.30 \times 10^2$ |
| MiddlePhalanxOutlineCorrect | 0.77 | $1.01 \times 10^6$ | $1.02 \times 10^6$ |
| EOGVerticalSignal | 0.77 | $6.38 \times 10^3$ | $5.62 \times 10^3$ |
| Chinatown | 0.84 | $1.71 \times 10^3$ | $2.05 \times 10^3$ |
| PLAID | 0.85 | $3.50 \times 10^2$ | $3.38 \times 10^2$ |
| ProximalPhalanxOutlineCorrect | 0.87 | $1.34 \times 10^1$ | $1.48 \times 10^1$ |
| EthanolLevel | 0.87 | $3.18 \times 10^1$ | $2.10 \times 10^1$ |
| Wine | 0.87 | $3.34 \times 10^4$ | $3.33 \times 10^4$ |
| Trace | 0.88 | $4.46 \times 10^3$ | $4.41 \times 10^3$ |
| ScreenType | 0.88 | $2.18 \times 10^2$ | $2.46 \times 10^2$ |
| Worms | 0.89 | $1.13 \times 10^2$ | $1.00 \times 10^2$ |
| BeetleFly | 0.89 | $5.79 \times 10^1$ | $5.30 \times 10^1$ |
| GesturePebbleZ1 | 0.90 | $4.34 \times 10^0$ | $3.98 \times 10^0$ |
| OliveOil | 0.91 | $5.64 \times 10^0$ | $5.94 \times 10^0$ |
| Strawberry | 0.91 | $1.59 \times 10^2$ | $1.56 \times 10^2$ |
| WormsTwoClass | 0.93 | $4.09 \times 10^1$ | $4.26 \times 10^1$ |
| Lightning7 | 0.94 | $3.32 \times 10^1$ | $3.80 \times 10^1$ |
| Meat | 0.94 | $2.80 \times 10^3$ | $1.35 \times 10^3$ |
| Plane | 0.94 | $9.58 \times 10^1$ | $1.01 \times 10^2$ |
| Beef | 0.94 | $6.40 \times 10^1$ | $6.78 \times 10^1$ |
| ProximalPhalanxOutlineAgeGroup | 0.94 | $4.70 \times 10^2$ | $7.09 \times 10^2$ |
| ShapesAll | 0.94 | $4.40 \times 10^1$ | $3.95 \times 10^1$ |
| ProximalPhalanxTW | 0.94 | $1.39 \times 10^4$ | $1.36 \times 10^4$ |
| MiddlePhalanxTW | 0.94 | $4.74 \times 10^0$ | $5.02 \times 10^0$ |
| SemgHandSubjectCh2 | 0.95 | $5.14 \times 10^1$ | $5.28 \times 10^1$ |
| ItalyPowerDemand | 0.95 | $2.75 \times 10^0$ | $2.92 \times 10^0$ |
| PhalangesOutlinesCorrect | 0.95 | $2.02 \times 10^1$ | $2.00 \times 10^1$ |
| DistalPhalanxOutlineCorrect | 0.96 | $5.31 \times 10^0$ | $6.94 \times 10^0$ |
| MoteStrain | 0.96 | $3.27 \times 10^1$ | $2.60 \times 10^1$ |
| CricketY | 0.96 | $3.90 \times 10^2$ | $3.94 \times 10^2$ |
| AllGestureWiimoteY | 0.96 | $1.57 \times 10^1$ | $1.63 \times 10^1$ |
| SwedishLeaf | 0.96 | $4.69 \times 10^2$ | $4.37 \times 10^2$ |
| ACSF1 | 0.96 | $1.01 \times 10^3$ | $1.04 \times 10^3$ |
| FaceAll | 0.97 | $3.58 \times 10^1$ | $3.67 \times 10^1$ |
| SemgHandGenderCh2 | 0.97 | $1.47 \times 10^2$ | $1.53 \times 10^2$ |
| DodgerLoopDay | 0.97 | $6.13 \times 10^2$ | $6.62 \times 10^2$ |
| NonInvasiveFetalECGThorax2 | 0.97 | $2.52 \times 10^0$ | $2.42 \times 10^0$ |
| Computers | 0.97 | $1.94 \times 10^2$ | $1.98 \times 10^2$ |
| MelbournePedestrian | 0.97 | $7.90 \times 10^1$ | $7.41 \times 10^1$ |
| AllGestureWiimoteX | 0.97 | $1.63 \times 10^2$ | $1.64 \times 10^2$ |
| UMD | 0.97 | $1.89 \times 10^1$ | $1.89 \times 10^1$ |
| ToeSegmentation2 | 0.97 | $2.03 \times 10^2$ | $1.72 \times 10^2$ |
| MixedShapesRegularTrain | 0.98 | $4.20 \times 10^2$ | $4.76 \times 10^2$ |
| OSULeaf | 0.98 | $8.85 \times 10^3$ | $6.43 \times 10^3$ |
| NonInvasiveFetalECGThorax1 | 0.98 | $1.31 \times 10^2$ | $1.33 \times 10^2$ |
| FordB | 0.98 | $2.81 \times 10^0$ | $2.80 \times 10^0$ |
| SmallKitchenAppliances | 0.99 | $2.49 \times 10^1$ | $2.61 \times 10^1$ |

| Dataset | Ratio | $Dispersion_{TEST}$ | $Dispersion_{TRAIN}$ |
|---|---|---|---|
| FordA | 0.99 | $3.73 \times 10^3$ | $3.83 \times 10^3$ |
| CricketZ | 0.99 | $2.55 \times 10^1$ | $2.52 \times 10^1$ |
| HouseTwenty | 0.99 | $2.44 \times 10^0$ | $2.79 \times 10^0$ |
| SemgHandMovementCh2 | 1.00 | $1.23 \times 10^4$ | $1.24 \times 10^4$ |
| CricketX | 1.00 | $6.78 \times 10^1$ | $6.10 \times 10^1$ |
| Earthquakes | 1.00 | $1.31 \times 10^2$ | $1.24 \times 10^2$ |
| TwoLeadECG | 1.00 | $2.28 \times 10^1$ | $2.32 \times 10^1$ |
| SonyAIBORobotSurface1 | 1.00 | $8.36 \times 10^0$ | $8.36 \times 10^0$ |
| MedicalImages | 1.00 | $7.57 \times 10^1$ | $8.10 \times 10^1$ |
| TwoPatterns | 1.00 | $5.83 \times 10^2$ | $3.90 \times 10^2$ |
| Crop | 1.00 | $1.28 \times 10^4$ | $1.35 \times 10^4$ |
| Fish | 1.00 | $1.13 \times 10^3$ | $9.94 \times 10^2$ |
| GunPointAgeSpan | 1.00 | $5.50 \times 10^0$ | $4.90 \times 10^0$ |
| FreezerRegularTrain | 1.01 | $2.47 \times 10^3$ | $3.27 \times 10^3$ |
| Herring | 1.01 | $1.02 \times 10^1$ | $1.07 \times 10^1$ |
| GestureMidAirD2 | 1.01 | $6.39 \times 10^0$ | $6.13 \times 10^0$ |
| ECGFiveDays | 1.01 | $5.42 \times 10^1$ | $4.85 \times 10^1$ |
| LargeKitchenAppliances | 1.01 | $3.68 \times 10^1$ | $3.08 \times 10^1$ |
| GunPointMaleVersusFemale | 1.02 | $3.69 \times 10^1$ | $5.17 \times 10^1$ |
| GunPointOldVersusYoung | 1.02 | $5.70 \times 10^2$ | $6.35 \times 10^2$ |
| Lightning2 | 1.02 | $5.96 \times 10^1$ | $1.31 \times 10^2$ |
| Yoga | 1.02 | $3.02 \times 10^4$ | $2.97 \times 10^4$ |
| AllGestureWiimoteZ | 1.02 | $1.06 \times 10^1$ | $9.93 \times 10^0$ |
| PowerCons | 1.02 | $2.07 \times 10^4$ | $1.63 \times 10^4$ |
| SyntheticControl | 1.02 | $2.29 \times 10^2$ | $1.92 \times 10^2$ |
| UWaveGestureLibraryX | 1.02 | $6.81 \times 10^1$ | $6.67 \times 10^1$ |
| GunPoint | 1.04 | $3.83 \times 10^2$ | $3.91 \times 10^2$ |
| UWaveGestureLibraryAll | 1.04 | $5.73 \times 10^1$ | $5.46 \times 10^1$ |
| FaceFour | 1.04 | $5.44 \times 10^1$ | $5.14 \times 10^1$ |
| DistalPhalanxTW | 1.04 | $2.07 \times 10^1$ | $2.07 \times 10^1$ |
| SmoothSubspace | 1.04 | $4.86 \times 10^1$ | $3.19 \times 10^1$ |
| UWaveGestureLibraryY | 1.05 | $2.00 \times 10^1$ | $1.73 \times 10^1$ |
| FiftyWords | 1.05 | $3.80 \times 10^0$ | $4.03 \times 10^0$ |
| StarLightCurves | 1.05 | $5.40 \times 10^4$ | $4.59 \times 10^4$ |
| ChlorineConcentration | 1.05 | $9.02 \times 10^1$ | $9.00 \times 10^1$ |
| RefrigerationDevices | 1.05 | $4.23 \times 10^1$ | $4.01 \times 10^1$ |
| UWaveGestureLibraryZ | 1.06 | $8.64 \times 10^0$ | $9.18 \times 10^0$ |
| InsectWingbeatSound | 1.06 | $7.54 \times 10^2$ | $7.85 \times 10^2$ |
| Coffee | 1.07 | $8.05 \times 10^0$ | $8.45 \times 10^0$ |
| Ham | 1.07 | $4.23 \times 10^2$ | $3.75 \times 10^2$ |
| InlineSkate | 1.07 | $8.25 \times 10^0$ | $6.80 \times 10^0$ |
| Haptics | 1.08 | $3.27 \times 10^1$ | $2.98 \times 10^1$ |
| Adiac | 1.09 | $2.81 \times 10^1$ | $2.25 \times 10^1$ |
| CBF | 1.09 | $6.69 \times 10^4$ | $8.63 \times 10^4$ |
| InsectEPGSmallTrain | 1.10 | $1.63 \times 10^2$ | $1.64 \times 10^2$ |
| ElectricDevices | 1.10 | $1.02 \times 10^2$ | $9.84 \times 10^1$ |
| DodgerLoopGame | 1.10 | $6.43 \times 10^2$ | $6.10 \times 10^2$ |
| WordSynonyms | 1.11 | $4.32 \times 10^3$ | $5.08 \times 10^3$ |
| FreezerSmallTrain | 1.11 | $2.29 \times 10^2$ | $2.35 \times 10^2$ |
| Mallat | 1.11 | $2.40 \times 10^1$ | $2.32 \times 10^1$ |

| Dataset | Ratio | $Dispersion_{TEST}$ | $Dispersion_{TRAIN}$ |
|---|---|---|---|
| FacesUCR | 1.12 | $1.20 \times 10^3$ | $1.08 \times 10^3$ |
| MiddlePhalanxOutlineAgeGroup | 1.12 | $2.70 \times 10^1$ | $2.24 \times 10^1$ |
| Wafer | 1.12 | $2.24 \times 10^2$ | $2.30 \times 10^2$ |
| ShapeletSim | 1.14 | $1.41 \times 10^4$ | $1.46 \times 10^4$ |
| ArrowHead | 1.16 | $1.71 \times 10^0$ | $1.88 \times 10^0$ |
| EOGHorizontalSignal | 1.18 | $3.01 \times 10^1$ | $2.65 \times 10^1$ |
| ToeSegmentation1 | 1.18 | $2.19 \times 10^2$ | $2.16 \times 10^2$ |
| SonyAIBORobotSurface2 | 1.18 | $2.80 \times 10^1$ | $2.36 \times 10^1$ |
| MixedShapesSmallTrain | 1.19 | $1.59 \times 10^2$ | $1.55 \times 10^2$ |
| ECG5000 | 1.19 | $4.17 \times 10^1$ | $4.77 \times 10^1$ |
| ECG200 | 1.21 | $1.28 \times 10^2$ | $1.25 \times 10^2$ |
| DistalPhalanxOutlineAgeGroup | 1.21 | $6.78 \times 10^1$ | $6.71 \times 10^1$ |
| CinCECGTorso | 1.24 | $1.41 \times 10^1$ | $1.40 \times 10^1$ |
| PickupGestureWiimoteZ | 1.25 | $5.23 \times 10^0$ | $5.98 \times 10^0$ |
| InsectEPGRegularTrain | 1.26 | $1.88 \times 10^1$ | $1.94 \times 10^1$ |
| Rock | 1.27 | $1.16 \times 10^2$ | $1.11 \times 10^2$ |
| BirdChicken | 1.30 | $5.28 \times 10^1$ | $5.47 \times 10^1$ |
| PigArtPressure | 1.38 | $1.03 \times 10^2$ | $9.85 \times 10^1$ |
| Phoneme | 1.50 | $5.18 \times 10^1$ | $4.70 \times 10^1$ |
| Car | 1.51 | $3.94 \times 10^2$ | $3.95 \times 10^2$ |
| PigCVP | 1.52 | $6.68 \times 10^1$ | $6.54 \times 10^1$ |
| Symbols | 1.53 | $1.23 \times 10^1$ | $3.72 \times 10^0$ |
| PigAirwayPressure | 2.07 | $7.11 \times 10^2$ | $5.72 \times 10^2$ |
| DiatomSizeReduction | 3.30 | $1.52 \times 10^3$ | $1.00 \times 10^3$ |

additional features (denoted by F23 and F24 below) considered in this study.

**F1: DN_HistogramMode_5** Top z-score range based on the highest count from a 5-bin histogram, representing the most frequent distribution range in the dataset.

**F2: DN_HistogramMode_10** Similar to DN5, but this considers the top z-score range based on a 10-bin histogram, providing a finer resolution.

**F3: CO_f1ecac** Represents the first 1/e crossing of the autocorrelation function, indicating how quickly the autocorrelation of a time series decays.

**F4: CO_FirstMin_ac** Identifies the first minimum of the autocorrelation function, which helps analyze the periodicity of the time series.

**F5: CO_HistogramAMI_even_2_5** Automutual information for $m = 2$ and $\tau = 5$, capturing the dependency between data points across time.

**F6: CO_trev_1_num** This statistic measures time-reversibility, focusing on the differences between successive points in the time series raised to the third power.

**F7: MD_hrv_classic_pnn40** Proportion of successive differences in time series values that exceed 0.04 of the standard deviation, indicating rapid fluctuations.

**F8: SB_BinaryStats_mean_longstretch1** The longest period where values stay consecutively above the mean, representing persistent trends in the data.

**F9: SB_TransitionMatrix_3ac_sumdiagcov** Trace of the covariance of the transition matrix between symbols in a 3-letter alphabet, used to assess transitions in symbolized data.

**F10: PD_PeriodicityWang_th0_01** A periodicity measure, indicating how regularly patterns repeat within the time series.

**F11: CO_Embed2_Dist_tau_d_expfit_meandiff** Exponential fit to the differences in distances between successive points in a 2-dimensional embedding space, revealing structural relationships.

**F12: IN_AutoMutualInfoStats_40_gaussian_fmmi** First minimum of the automutual information function, which gives insight into the periodicity and structure of the time series.

**F13: FC_LocalSimple_mean1_tauresrat** Measures the change in correlation length after

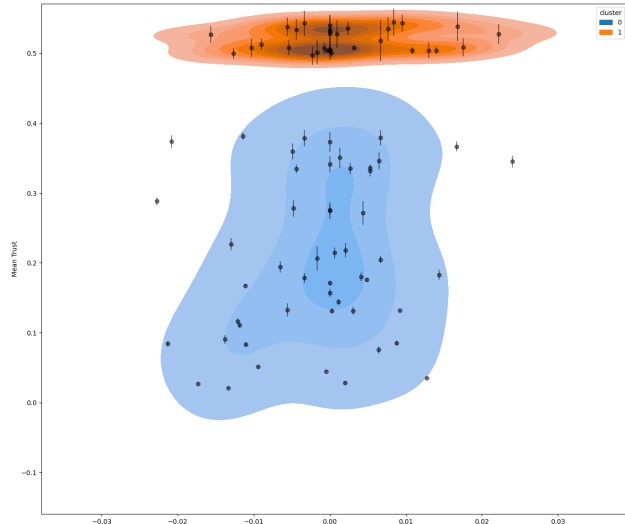

Figure 19: **Clustering of the confidence in regards to performance for FCN:** Overview of the relationship between mean confidence over the augmentation steps and final performance through a DBSCAN clustering.

.

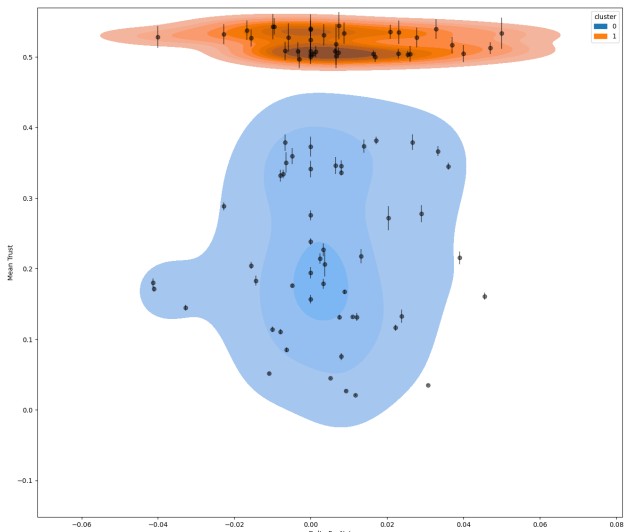

Figure 20: **Clustering of the confidence in regards to performance for ResNet:** Overview of the relationship between mean confidence over the augmentation steps and final performance through a DBSCAN clustering.

.

iteratively differencing the time series, providing insights into the stationarity of the data.

**F14:** `DN_OutlierInclude_p_001_mdrmd` Measures the time intervals between successive extreme events occurring above the mean, indicating patterns of high values.

**F15:** `DN_OutlierInclude_n_001_mdrmd` Similar to DNOp but for extreme events occurring below the mean, highlighting the time intervals between low-value outliers.

**F16:** `SP_Summaries_welch_rect_area_5_1` This computes the total power in the lowest fifth of the frequencies from a Fourier power spectrum, reflecting long-term trends.

**F17:** `SB_BinaryStats_diff_longstretch0` The longest period of successive decreases in the time series, capturing prolonged declining trends.

**F18:** `SB_MotifThree_quantile_hh` Shannon entropy of successive symbol pairs in a 3-letter quantile symbolization, quantifying the complexity of transitions between motifs.

**F19:** `SC_FluctAnal_2_rsrangefit_50_1_logi_prop_r1` Proportion of slower timescale fluctuations that scale with rescaled range fits, indicating long-term memory in the data.

**F20:** `SC_FluctAnal_2_dfa_50_1_2_logi_prop_r1` Proportion of slower timescale fluctuations that scale with detrended fluctuation analysis (DFA) under 50

**F21:** `SP_Summaries_welch_rect_centroid` The centroid of the Fourier power spectrum, which offers a measure of the central frequency or the dominant pattern in the time series.

**F22:** `FC_LocalSimple_mean3_stderr` Calculates the mean error from a rolling 3-sample mean forecast, capturing the volatility of short-term predictions.

**F23:** `Train_Test_Ratio` The ratio of training data to test data in the dataset.

**F24:** `Discrepancy_in_Distance` To estimate the discrepancy in distance between the training and testing set distributions, as defined in Appendix E.1

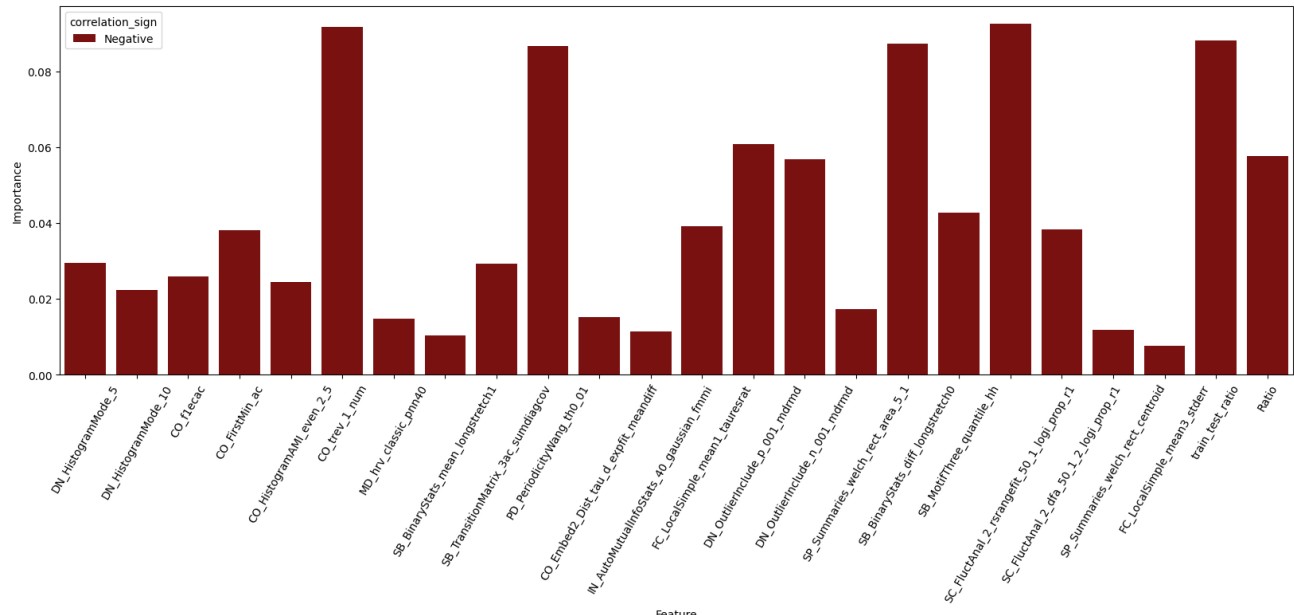

Figure 21: **Feature importance in regards to confidence:** Overview of the impact of every dataset feature on the mean confidence over the augmentation steps. The red color denotes the negative correlation these features hold with confidence.
.

# G. Evolution of latent space through learning phase

A progressive visualization of the latent space offers valuable insights into the evolving distribution modeling and exploration process. Initially, the latent space representations exhibit fine clustering, but as we iterate in the augmentation loop, the latent space distributions become denser, enhancing the exploration part of these distributions. However, in the later stages of augmentation, the exploration process becomes increasingly challenging as the inter-class distances appear to shrink due to prior augmentation steps. It is important to note that these visualizations provide only a limited view of the actual distributions, as they are restricted to three dimensions (from an original 50-dimensional space).

Table 8: Latent Space Evolution. Visualization of the latent space for the 3 first dimensions (out of 50)

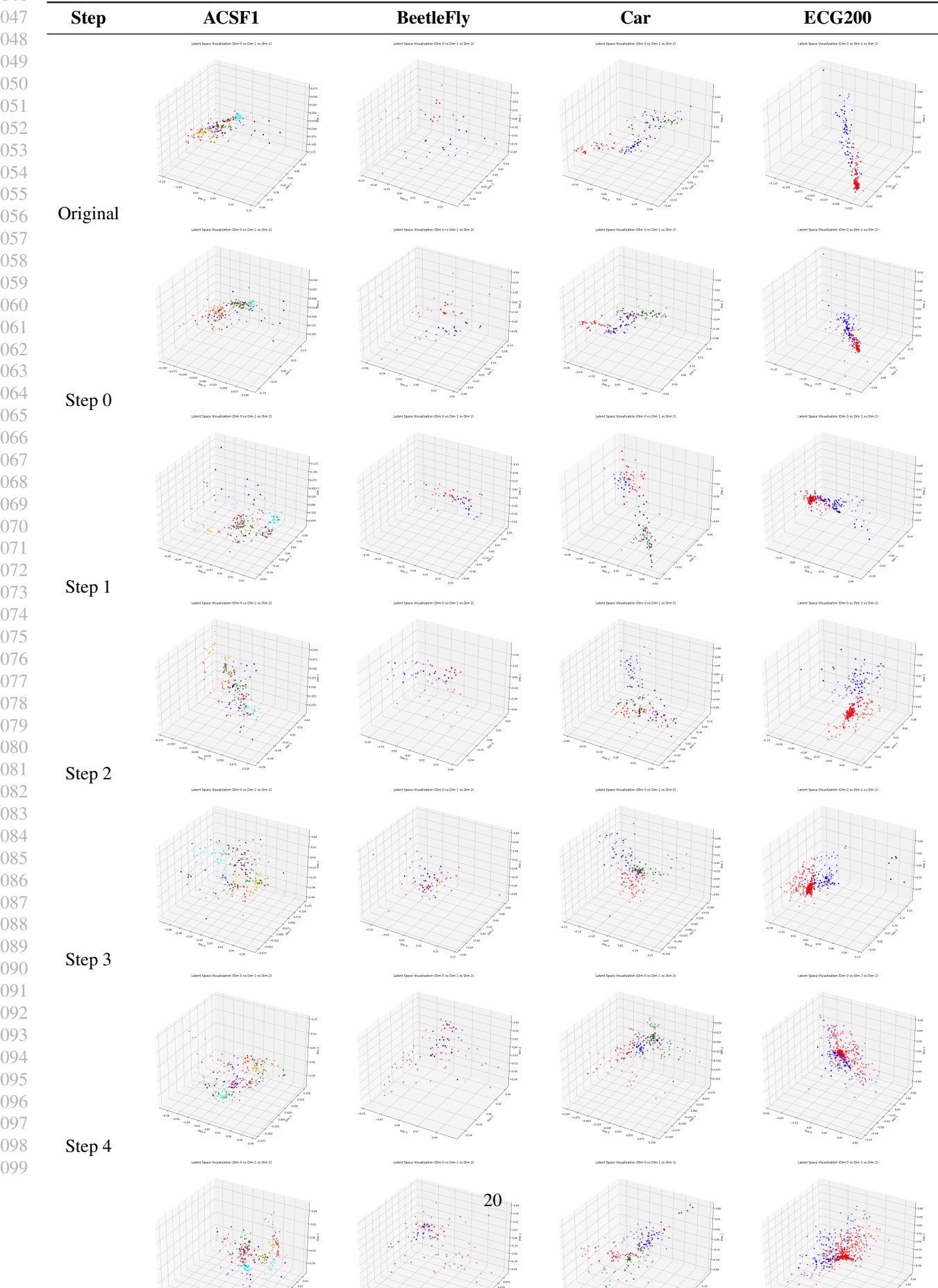

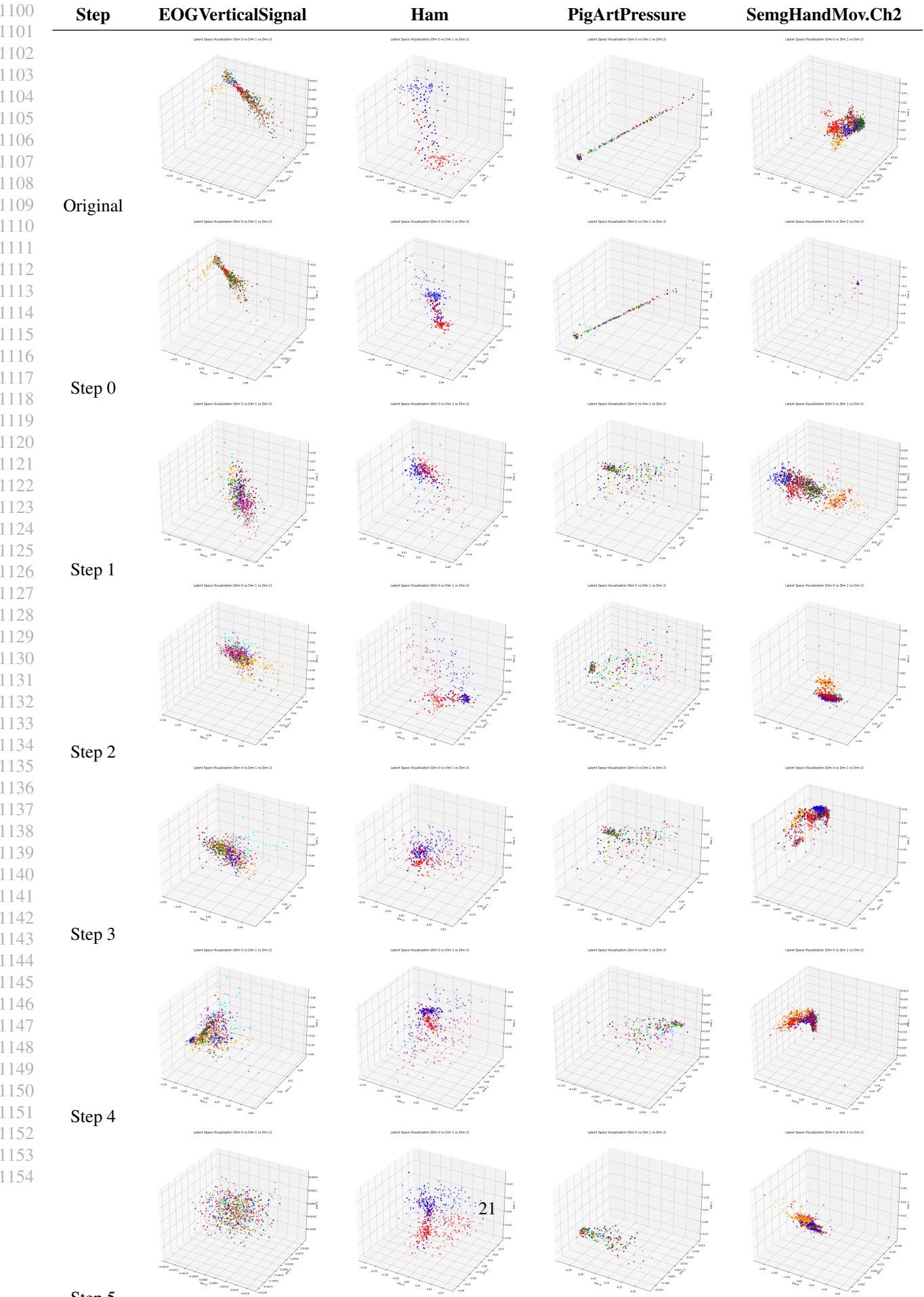

| Step | UMD | Worms |
|------|-----|-------|

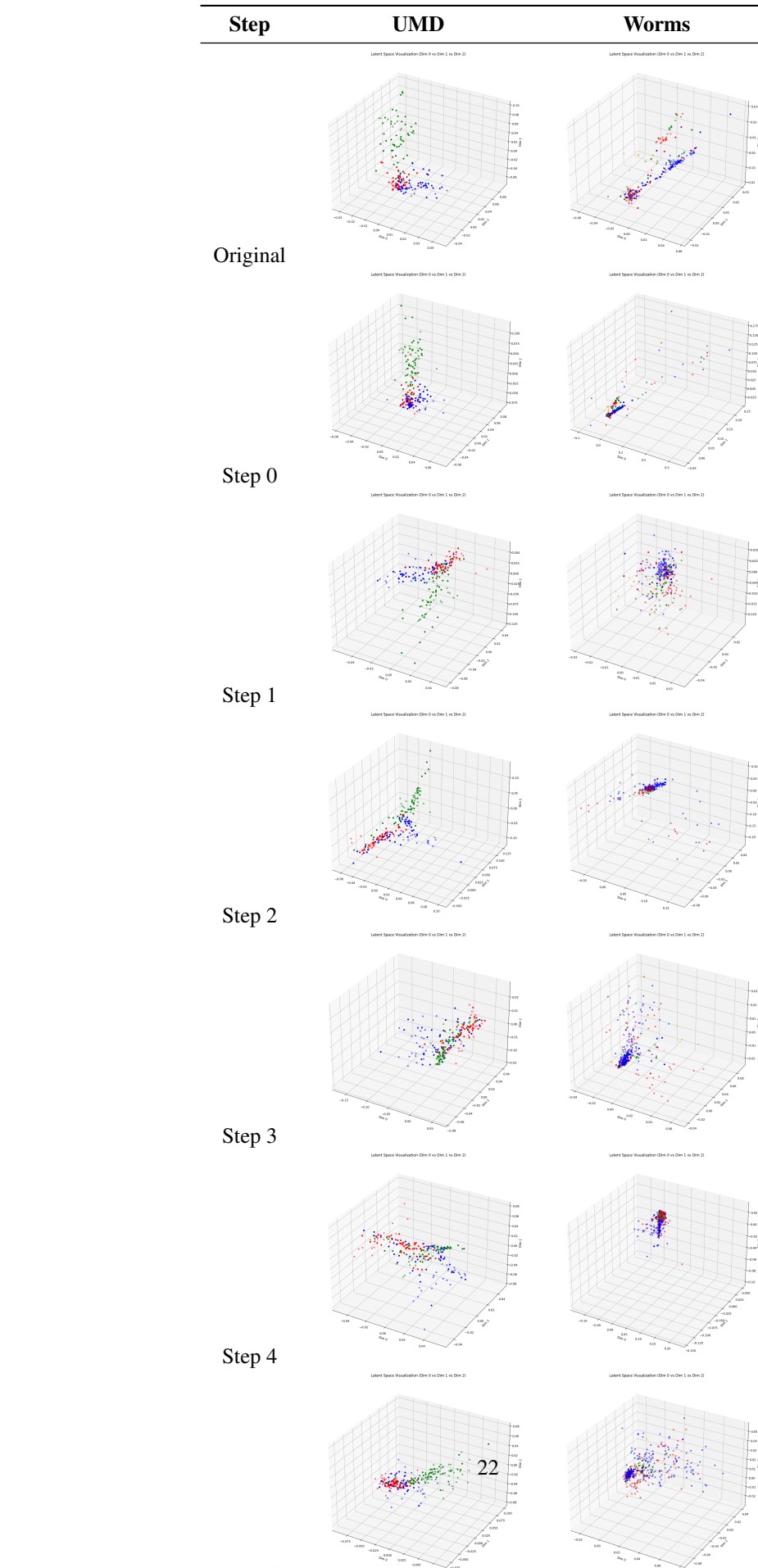

Original

Step 0

Step 1

Step 2

Step 3

Step 4

22

Step 5

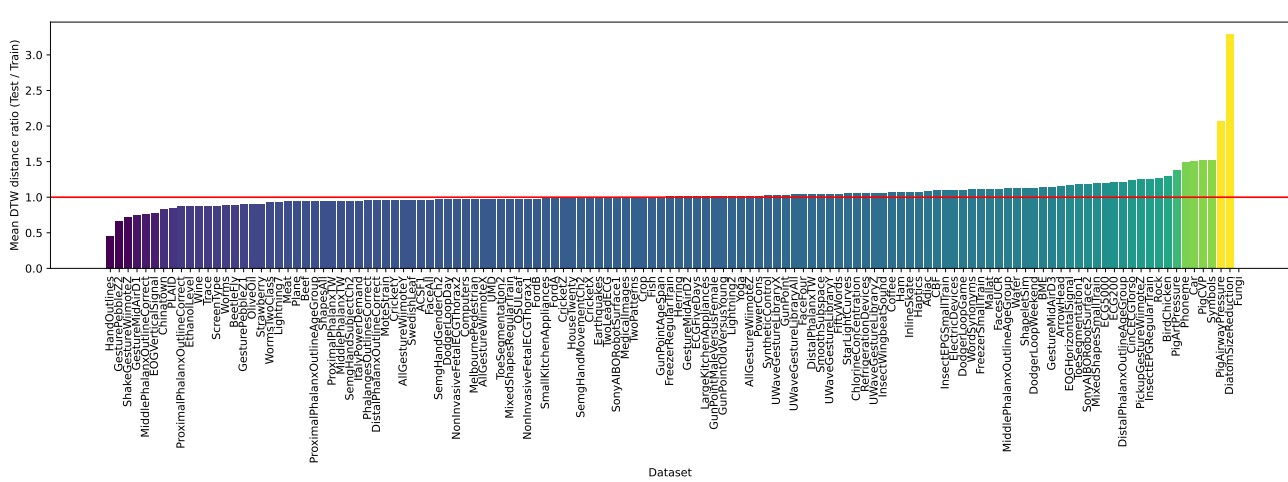

Figure 22: **Distribution discrepancy ratio:** Overview of the difference in discrepancy between training and testing sets of the 102 UCR datasets; discrepancy ratio computed using (8)

.

