# OpenReview forum: "ASCENSION: Autoencoder-Based Latent Space Class Expansion for Time Series Data Augmentation"
_ICML.cc/2025/Conference — Submitted to ICML 2025_

### Official Review · Reviewer_X9XC · 2025-03-12

**Overall Recommendation:** 3

**Summary:**

A summary of the paper: This paper presents ASCENSION, a VAE-based data augmentation framework designed to address distribution discrepancies in Class Expansion. It uses latent space to improve the applicability of data augmentation and evaluates ASCENSION’s impact on classification performance across various time-series datasets.
Key weaknesses:
1.The claimed novelty of the paper lies in latent space and the use of VAE for data augmentation in time-series, but these techniques are not fundamentally new. Latent space clustering, in particular, is well-documented and widely used in existing literature. Additionally, the paper lacks a detailed discussion of the specific limitations in time-series scenarios and how ASCENSION addresses class expansion in these contexts. The contributions seem to be a combination of existing methods, tested in a specific setting, without introducing significant new innovations.

2.The motivation of the paper is unclear. In the introduction, the authors mention issues like the lack of research on periodicity in data augmentation, VAE’s limitations in class expansion, and the divergence between training and operational performance, but the specific problem the paper aims to address remains confusing. As a result, the specific problem the paper aims to address remains unclear. A clearer definition of the problem would strengthen the paper’s focus.

3.The paper lacks detailed justification for its proposed method, which does not fully align with the stated contributions. For instance, the introduction mentions controlled and progressive expansion of class probability densities and boundaries, as well as preventing harmful overlap, but these concepts are not adequately explored or mathematically developed in the method section. This lack of detail diminishes the clarity and impact of the paper’s contributions.

4.The experimental section does not provide enough detail to demonstrate how ASCENSION effectively expands class probability densities and boundaries as claimed.

5.The paper lacks clarity and coherence, making it difficult to follow. For example, the phrase “demonstrate potential” (Line 17) is uncommon. In Line 42, the term “distribution discrepancy ratio” used to explain "when training and operational data distributions diverge" is redundant and still fails to pinpoint the underlying causes of the limitations. Additionally, key details are scattered across various sections, including the appendix, making it hard to follow the argument.

**Claims And Evidence:**

The claims of novelty are problematic because the use of latent space and VAE  is already well-documented. The paper also doesn’t clearly explain the specific limitations in time-series and how ASCENSION addresses class expansion.

**Essential References Not Discussed:**

The paper cites relevant related work on data augmentation but fails to clearly explain the limitations of these methods and how the proposed approach addresses them. The references are not fully used to frame the contributions or justify the motivation.

**Experimental Designs Or Analyses:**

The evaluation method in the paper focuses only on overall classification accuracy and does not provide enough detail to demonstrate how ASCENSION effectively expands class probability densities and boundaries as claimed.

**Methods And Evaluation Criteria:**

The evaluation method in the paper focuses only on overall classification accuracy and does not provide enough detail to demonstrate how ASCENSION effectively expands class probability densities and boundaries as claimed. Additionally, the baseline models used for comparison are not the most recent.

**Other Comments Or Suggestions:**

No

**Other Strengths And Weaknesses:**

No

**Questions For Authors:**

No

**Relation To Broader Scientific Literature:**

I believe the paper’s contributions have limited relation to the broader scientific literature. The techniques used, such as latent space clustering and VAE for data augmentation, are not new and have been widely explored in prior work.

**Theoretical Claims:**

The proposed method is relatively simple and primarily descriptive, lacking detailed proofs or a deeper theoretical explanation to support its claims.

---

> ### Author Rebuttal · Authors · 2025-04-01
>
> Thank you for the review. Below, we clarify the novelty and motivation of our work and provide new results supporting our key hypotheses
>
> ## Summary
> "**C4.1** *Claimed novelty lies in latent space and use of VAE for DA in time-series, but these are not fundamentally new **C4.2** Motivation of the paper is unclear **C4.3** Paper doesn’t clearly explain th limitations in time-series and how ASCENSION addresses class expansion.*"
>
> This work is motivated by the fact that most state-of-the-art DA methods for time series focus on intra-class generation. We hypothesize that controlled, progressive class boundary expansion in latent space can boost classification. While prior works (e.g., Modals, [Wa24, Wa24b]) explore class expansion, they do not include gradual control. Our key contribution, the α-scaling mechanism, enables this, going beyond clustering loss combination. Experiments show ASCENSION consistently outperforms baselines: with ResNet, gains of 2.8%–8.5%, and up to 30.2% vs. KoVAE; with FCN, 1.3%–13.2%, and up to 31.7%. A new ablation study (added to rebuttal) attributes 7%–61% of the gains to α-scaling. Due to space limits, we refer the reviewer to our response to Reviewer h2Y9 (C3.5) for more details about this ablation study. These new results will be included in the revised paper.
>
> [Wa24] Wang, T. et al. (2024) Fine-grained Control of Generative Data Augmentation in IoT Sensing. Advances in Neural Information Processing Systems, 37, 32787-32812.
>
> [Wa24b] Wang, T. et al. (2024) Data augmentation for human activity recognition via condition space interpolation within a generative model. ICCCN (2024)
>
> "**C4.4** *Paper lacks clarity and coherence; Key details are scattered across sections, making it hard to follow...*"
>
> Thank you for pointing out the language issues - we’ve corrected them. We’re also open to any specific suggestions the reviewer may have regarding content organization."
>
> ## Methods And Evaluation Criteria
>
> "**C4.5** *Evaluation method focuses only on overall classification accuracy and does not provide enough detail to demonstrate how ASCENSION effectively expands class probability densities/boundaries. Baseline models are not the most recent*"
>
> Section 4.2.6 and Fig. 5 qualitatively assess class expansion risks, with quantitative details in Appendix D. Based on reviewer suggestions, we added ImagenTime, Diffusion-TS, and KoVAE to our baselines. ASCENSION outperforms all: for FCN, they rank 3rd, 5th, and 9th  in total accuracy; for ResNet, 4th, 6th, and 9th. Summary results appear below; Full results available at: https://github.com/ASCENSION-PAPER/ASCENSION/tree/main/comparison_results
>
> - ResNet
>
> | Method       | ↑Nb Augmented | ↑Augmented mean acc | Nb Unchanged | Unchanged mean acc | ↓Nb Worsened | ↑Worsened mean acc | Nb Total | ↑Total mean acc |
> |--------------|---------------|---------------------|--------------|--------------------|--------------|--------------------|----------|-----------------|
> | ASCENSION    | **56** | **4.0**|16 | 0.0 |**30** | -**1.7**| 102 | **1.7** |
> | ImagenTime   | 26 | 1.8 |17 | 0.0  |59 | -6.2 | 102 | -3.1 |
> | Diffusion-TS | 30 | 1.3| 6 | 0.0 |66 | -9.2 | 102 | -5.5|
> | KoVAE        | 1 | 0.7 |6 | 0.0 |95 | -30.6| 102 | -28.5|
>
> - FCN
>
> | Method       | ↑Nb Augmented | ↑Augmented mean acc | Nb Unchanged | Unchanged mean acc | ↓Nb Worsened | ↑Worsened mean acc | Nb Total | ↑Total mean acc |
> |--------------|---------------|---------------------|--------------|--------------------|--------------|--------------------|----------|-----------------|
> | ASCENSION    | **50** | **3.0** |13 | 0.0 |**39** | -**1.4** | 102 | **1.0**|
> | ImagenTime   | 25 | 2.8 |13 | 0.0| 64 | -3.0 | 102 | -1.2 |
> | Diffusion-TS |37 | **3.0** | 7 | 0.0|58 | -14.8| 102 | -7.3 |
> | KoVAE        | 3 | 7.1| 2 | 0.0|97 | -32.5 | 102 | -30.7|
>
> ## Theoretical Claims
> "**C4.6** *The method lacks detailed proofs or a deeper theoretical explanation to support its claims.*"
>
> Despite its simplicity, extensive empirical comparisons substantiate our method’s effectiveness against SoTa alternatives. The new ablation study quantifies the α-scaling mechanism and contrastive loss impact (cf response to Reviewer h2Y9 - C3.5). Although contrastive loss benefits are established (>2% accuracy increase), notably, the α-scaling mechanism contributes significantly (7 to 44%) of accuracy improvement over ResNet and 10 to 61% over FCN.
>
> Existing sections will be refined to enhance clarity of the justification."
>
> ## Supplementary Material
>
> "**C4.7** *No. I didn't find the materials.*"
>
> Supplementary materials are available on our anonymous GitHub (see Sec. 6): https://github.com/ASCENSION-PAPER
>
> ## Relation To Broader Scientific Literature
> "**C4.9** *Paper fails to explain limitations of SoTa methods and how ASCENSION addresses them.*"
>
> Appendix A outlines key limitations (e.g., temporal distortion, GAN instability) and explains how ASCENSION advances the state of the art. Figure 6 gives a timeline of all baselines.

---

### Official Review · Reviewer_h2Y9 · 2025-03-14

**Overall Recommendation:** 4

**Summary:**

This paper introduces a VAE-based generative data augmentation approach for time-series data called ASCENSION. This work aims at progressively expanding inter-class boundary during the generation, enabling the exploration of underrepresented or unseen latent distribution in the training data. The major technical innovation lies in the design of clustering loss and iterative training process. Comprehensive experiments are conducted to demonstrate the effectiveness of ASCENSION.

**Claims And Evidence:**

The experimental results presented in this paper provide strong support for the claims made. However, the assertion that "To our knowledge, no state-of-the-art DA method for time-series classification enables progressive (iterative) and meaningful class boundary expansion during synthetic data generation" may require reconsideration. The work presented in [1] also appears to propose a progressive class boundary expansion for time-series generative data generation. While the methodologies differ—ASCENSION manipulates features in the latent space, whereas [1] focuses on controlling conditions—it would be beneficial to compare these two approaches and revise the original claim accordingly. This comparison would provide a more comprehensive and accurate representation of the current state of the art in this field.

[1] Fine-grained Control of Generative Data Augmentation in IoT Sensing, NeurIPS 2024

**Essential References Not Discussed:**

Please see **Relation To Broader Scientific Literature**.

**Experimental Designs Or Analyses:**

The experimental work presented in this paper is comprehensive. However, a significant issue is the absence of an ablation study. Such a study would be particularly valuable in demonstrating the individual impacts of two key components: the clustering loss and the iterative training approach. Specifically, it would be beneficial to understand how each of these elements independently contributes to the overall performance of the proposed method.

**Methods And Evaluation Criteria:**

The proposed approach of incorporating clustering loss into VAE training appears intuitive and straightforward. However, the implementation details are not clearly outlined. Specifically, how is the clustering loss computed during training? Is the distance loss calculated for each data point within a batch? How does this approach compare to or differ from contrastive learning? Additionally, how does the weighting of the loss terms impact the final performance?

The paper also introduces an iterative training approach, but the rationale behind this choice is not clearly explained. A more straightforward alternative could be to control class expansion by simply adjusting the clustering loss. The authors' own analysis at Line 651 suggests that the iterative training method is prone to instability and may lead to error accumulation over time. This instability raises concerns about the practical applicability of the technique. To strengthen their case, the authors should provide a more comprehensive justification for the iterative training approach. Specifically, they need to explain its fundamental advantages over other potential methods. A comparative analysis demonstrating why this approach was selected over seemingly simpler and potentially more stable alternatives would greatly enhance the credibility and value of their proposed method.

**Other Comments Or Suggestions:**

The paper is well-written and easy to follow. There is no apparent typo or formatting issue that I noticed.

**Other Strengths And Weaknesses:**

This paper proposed an intuitive improvement over VAE for time-series data augmentation. However, there remains ambiguity regarding the implementation of the method that requires further clarification. The motivation of iterative training also stays unclear. A large amount of experiments are conducted to show that ASCENSION's comparative performance over the baselines in various TSC tasks. But ablation studies are required in order to prove the validity of the design choices.

**Questions For Authors:**

The paper exhibits strong merit through its thorough comparative experiments. Its findings will likely serve as valuable reference points for researchers in the field. While I have raised several questions in my previous comments, particularly regarding the iterative training justification, progressive boundary expansion claims, and the need for ablation studies, I remain positive about the overall contribution. I would be inclined to provide a higher rating if the authors can address these concerns with reasonable explanations.

**Relation To Broader Scientific Literature:**

As mentioned in **Claims And Evidence**, prior research has explored the potential of generating inter class synthetic samples as a data augmentation approach [1][2]. It would be beneficial to do a wider survey and incorporate the findings into the related work.

[1] Fine-grained Control of Generative Data Augmentation in IoT Sensing
[2] Data augmentation for human activity recognition via condition space interpolation within a generative model

**Theoretical Claims:**

No theoretical claims are made in this paper.

---

> ### Author Rebuttal · Authors · 2025-04-01
>
> We appreciate the reviewer’s detailed and constructive comments. We address each of the raised concerns below. We clarify the novelty and rationale of ASCENSION compared to the referenced works below. We also provide ablation study results.
>
> ## Claims And Evidence & Relation To Broader Scientific Literature
> "**C3.1** *the assertion that 'no SoTa DA method for time-series classification enables progressive (iterative) and meaningful class boundary expansion...' may require reconsideration. [1] also appears to propose a progressive class boundary expansion...
> prior research has explored the potential of generating inter class synthetic samples [1,2]. It would be beneficial to do a wider survey and incorporate the findings into the related work.*"
>
> References [1, 2] explore class boundary expansion via inter-class interpolation, but our approach differs in two key ways: (i) we extrapolate within a single class for controlled expansion, and (ii) the process is iterative and progressive—*starting small and expanding gradually (cf. Fig. 1)* —while assessing risk during sample generation, eliminating the need to assign class labels. We thank the reviewer for pointing out these works and will include them in the revised Related Work (App. A). Unfortunately, as public implementations are not yet available, we could not include them in our comparison. Nonetheless, we added three new DA baseline methods—ImagenTime, DiffusionTS, and KoVAE—as suggested by Reviewer SfpM. Full version of results available at: https://github.com/ASCENSION-PAPER/ASCENSION/tree/main/comparison_results
>
> ## Methods And Evaluation Criteria
> "**C3.2** *how is the clustering loss computed during training? Is the distance loss calculated for each data point within a batch? how does the weighting of the loss terms impact the final performance?*"
>
> $\mathcal{L}_{cluster}$ is a contrastive loss computed per batch and back-propagated with the total loss via SGD. We initially used uniform weights, but following the reviewer’s suggestion, we are running a grid search over 256 combinations. So far, 2 have been tested, showing only a 0.02% accuracy variation—too early for conclusions, but more results will be included in the revised version and Supplementary Material.
>
> "**C3.3** *How does this approach compare to or differ from contrastive learning?*"
>
> We thank the reviewer for noting the imprecise description, $\mathcal{L}_{cluster}$ is in fact a contrastive loss and this will be clarified in the revised version.
>
> "**C3.4** *The rationale behind this (iterative training) choice is not clearly explained. A more straightforward alternative could be to control class expansion by simply adjusting the clustering loss. The authors' own analysis at Line 651 suggests that the iterative training method is prone to instability ... they need to explain its fundamental advantages over other potential methods*"
>
> The rationale for iterative training stems from our hypothesis that progressively and controllably expanding class boundaries in latent space improves classification. While prior works (e.g., Modals and [1]) have explored class expansion, they lack mechanisms for gradual control. Our α-scaling mechanism preserves clear class boundaries before extrapolating outside a class space—unlike approaches that simply reduce clustering loss early on. Despite some instability, ASCENSION shows consistent improvements across all UCR 102 datasets: with ResNet, gains range from 2.8%–8.5% (Table 1) and up to 30.2% vs. KoVAE; with FCN, 1.3%–13.2%, and up to 31.7% vs. KoVAE. Due to space limits, we refer the reviewer to the new results table in our response to Reviewer SfpM (see C2.2). An ablation study (detailed below) attributes 7%–61% of these improvements to progressive class expansion. These findings will be included in the revised version.
>
>  ## Experimental Designs Or Analyses
>
> "**C3.5** *A significant issue is the absence of an ablation study; it would be beneficial to understand how each of these elements independently contributes to the overall performance.*"
>
> An ablation study was conducted for the rebuttal to assess the individual roles of the clustering loss and α-scaling mechanism - full results at: https://github.com/ASCENSION-PAPER/ASCENSION/tree/main/ablation_study/
>
>  + the α-scaling mechanism significantly improves performance, with median gains of 0.3–0.5% and top-quartile gains over 1.9%, accounting for 7–44% (ResNet) and 10–61% (FCN) of the accuracy gap with baselines. On 102 datasets, the performance delta (mean accuracy step>1 – step=1) is:
>
> | Classifier | Q1 | Median | Q3 | IQR |
> |------------|--------|--------|--------|--------|
> | ResNet | 0.00000 | 0.00502 | 0.01923 | 0.01923 |
> | FCN | 0.00000 | 0.00333 | 0.01346 | 0.01346 |
>
>  + removing the clustering loss yields only a 1.3% averafe gain (vs. 4% with the full method), with benefits limited to the first augmentation step. Subsequent degradation highlights its key role in supporting progressive expansion

---

> > ### Comment · Reviewer_h2Y9 · 2025-04-05
> >
> > I appreciate the authors reply. I would like to raise my rating.

---

### Official Review · Reviewer_SfpM · 2025-03-15

**Overall Recommendation:** 2

**Summary:**

This paper introduces ASCENSION, a VAE-based data augmentation (DA) technique tailored for time series classification (TSC). The core idea centers on a controllable and progressive latent space class expansion mechanism, leveraging the structured latent space of VAEs. ASCENSION aims to overcome the limitations of traditional and generative DA methods—particularly class boundary rigidity and overfitting to the training distribution. The model introduces a clustering loss to ensure intra-class compactness and inter-class separability and iteratively expands latent distributions with a tunable α-scaling factor.
Empirical evaluations are conducted across 102 datasets from the UCR archive, comparing ASCENSION against DA baselines. Results show that ASCENSION achieves the most consistent classification gains with fewer instances of performance degradation.

**Claims And Evidence:**

The paper’s claims regarding robust performance gains, latent class expansion, and distributional generalization are somewhat supported by empirical evidence. However, there are concerns:

- Several notations and definitions (e.g., $L_{class}$, $K_y$) are poorly explained or introduced late, reducing clarity around the contribution and optimization objectives.
- The absence of fair comparisons with strong generative baselines (e.g., DiffusionTS, KoVAE, ImagenTime) limits the strength of the benchmarking.

**Essential References Not Discussed:**

The paper omits several recent generative time series models, DiffusionTS, GT-GAN, KoVAE, and ImagenTime, which offer stronger baselines for time series generation.

**Experimental Designs Or Analyses:**

The experimental setup is mainly sound, with multiple classifiers (ResNet/FCN) and diverse datasets. However:

- The use of $L_{class}$ loss is unexplained in Section 3 before its inclusion in the final loss formulation.
- It is unclear what $K_y$ represents in Equation (3)—the number of mixture components?
- Fairness of experimental baselines is questionable due to conditional generation mismatch.
- No ablations are presented to show the necessity of the clustering loss or the $\alpha$-scaling expansion mechanism independently.

**Methods And Evaluation Criteria:**

The evaluation across 102 datasets is impressive in scope. However, evaluation fairness is questionable, as there are state-of-the-art time series generation methods that are not compared, such as ImagenTime [1], DiffusionTS [2], KoVAE [3], and GT-GAN [4].

[1] Naiman, Ilan, et al. "Utilizing image transforms and diffusion models for generative modeling of short and long time series."

[2] Yuan, Xinyu, and Yan Qiao. "Diffusion-ts: Interpretable diffusion for general time series generation."

[3] Naiman, Ilan, et al. "Generative modeling of regular and irregular time series data via Koopman VAEs."‏

[4] Jeon, Jinsung, et al. "GT-GAN: General purpose time series synthesis with generative adversarial networks."

**Other Comments Or Suggestions:**

None

**Other Strengths And Weaknesses:**

N/A

**Questions For Authors:**

Why $L_{cluster}$ is used and not simply contrastive loss?

**Relation To Broader Scientific Literature:**

It aims to provide a robust data augmentation method.

**Theoretical Claims:**

No

---

> ### Author Rebuttal · Authors · 2025-04-01
>
> We thank the reviewer for the detailed and constructive critique. We provide the requested clarifications below and provide comparison results with the suggested baselines. We will include all this in our revised manuscript.
>
> ## Claims And Evidence & Methods And Evaluation Criteria
> "**C2.1** *Several notations and definitions ($\mathcal{L}_{class}$, $K_y$) are poorly explained, reducing clarity around the contribution*"
>
> To enhance clarity, these notations will be further explained in the revision. Specifically, $L_{class}$ denotes the contrastive loss, while $K_y$ represents the number of points for class $y$ at the current augmentation step. The paper will be reorganized so that $\mathcal{L}_{class}$ and $K_y$ are defined as soon as they become relevant.
>
> "**C2.2** *The absence of fair comparisons with strong generative baselines such as ImagenTime [1], DiffusionTS [2], KoVAE[3], GT-GAN [4]*"
>
> We have conducted additional experimental comparison studies for the rebuttal, incorporating the suggested methods. These new results will be incorporated into Section 4 (Results). Unfortunately, we were unable to reproduce GT-GAN, as the available code contains hard-coded parameters and would require substantial modifications. As it stands, the implementation appears difficult to reproduce without further clarification or updates.
>
> A summary of the new results is provided below, demonstrating that ASCENSION significantly outperforms the evaluated baselines. ImagenTime, Diffusion-TS, and KoVAE are respectively ranked -- in terms of "total" accuracy performance -- 3rd, 5th and 9th for FCN (among all the evaluated baselines), and 4th, 6th and 9th for ResNet.
>
> Full version of the new experiments (for all datasets) is available at: https://github.com/ASCENSION-PAPER/ASCENSION/tree/main/comparison_results
>
> - ResNet
>
> | Method       | ↑Nb Augmented | ↑Augmented mean acc | Nb Unchanged | Unchanged mean acc | ↓Nb Worsened | ↑Worsened mean acc | Nb Total | ↑Total mean acc |
> |--------------|---------------|---------------------|--------------|--------------------|--------------|--------------------|----------|-----------------|
> | ASCENSION    | **56** | **4.0**|16 | 0.0 |**30** | -**1.7**| 102 | **1.7** |
> | ImagenTime   | 26 | 1.8 |17 | 0.0  |59 | -6.2 | 102 | -3.1 |
> | Diffusion-TS | 30 | 1.3| 6 | 0.0 |66 | -9.2 | 102 | -5.5|
> | KoVAE        | 1 | 0.7 |6 | 0.0 |95 | -30.6| 102 | -28.5|
>
> - FCN
>
> | Method       | ↑Nb Augmented | ↑Augmented mean acc | Nb Unchanged | Unchanged mean acc | ↓Nb Worsened | ↑Worsened mean acc | Nb Total | ↑Total mean acc |
> |--------------|---------------|---------------------|--------------|--------------------|--------------|--------------------|----------|-----------------|
> | ASCENSION    | **50** | **3.0** |13 | 0.0 |**39** | -**1.4** | 102 | **1.0**|
> | ImagenTime   | 25 | 2.8 |13 | 0.0| 64 | -3.0 | 102 | -1.2 |
> | Diffusion-TS |37 | **3.0** | 7 | 0.0|58 | -14.8| 102 | -7.3 |
> | KoVAE        | 3 | 7.1| 2 | 0.0|97 | -32.5 | 102 | -30.7|
>
> ## Experimental Designs Or Analyses
> "**C2.3** *Fairness of experimental baselines is questionable due to conditional generation mismatch.*"
>
> We are not sure to understand what the reviewer means by "conditional generation mismatch". We would be happy to follow up on this topic if the reviewer can clarify.
>
> "**C2.4** *No ablations are presented to show the necessity of the clustering loss or the α-scaling expansion mechanism independently*"
>
> "Two ablation studies were conducted to assess the individual roles of the clustering loss and α-scaling mechanism. These new studies will be added to the revised version. Full results: https://github.com/ASCENSION-PAPER/ASCENSION/tree/main/ablation_study
>
> The results and findings of these two new studies are presented below:
> + the α-scaling mechanism significantly improves performance, with median gains of 0.3–0.5% and top-quartile gains over 1.9%, accounting for 7–44% (ResNet) and 10–61% (FCN) of the accuracy gap with baselines. On 102 datasets, the performance delta (mean accuracy step>1 – step=1) is:
>
> | Classifier | Q1 | Median | Q3 | IQR |
> |------------|--------|--------|--------|--------|
> | ResNet | 0.00000 | 0.00502 | 0.01923 | 0.01923 |
> | FCN | 0.00000 | 0.00333 | 0.01346 | 0.01346 |
>
> + Removing the clustering loss yields only a modest average accuracy gain of 1.3%, compared to 4% with the full method. Notably, improvements are mostly observed in the initial augmentation step, while later progressive steps lead to rapid accuracy degradation without the clustering loss, thus highlighting its essential role in sustaining performance throughout the augmentation process.
>
> "**C2.5** *Questions For Authors - Why $\mathcal{L}_{cluster}$ is used and not simply contrastive loss?*"
>
> The term $\mathcal{L}_{cluster}$ in our paper refers to a contrastive loss. We thank the reviewer for highlighting the lack of clarity and will revise the manuscript accordingly.

---

### Official Review · Reviewer_ZQER · 2025-03-18

**Overall Recommendation:** 4

**Summary:**

This work introduced a data augmentation method for time series data called ASCENSION. The method is based on the classic VAE-based training and sampling process, but the authors incorporated a clustering loss into training for better classification performance. The proposed method iteratively augments the dataset by repeating the training and sampling process multiple times. The proposed method is systematically evaluated and applied on UCR dataset, and show stable performance improvement over benchmarks.

**Claims And Evidence:**

The authors systematically evaluated their method on all UCR datasets, and demonstrated clear performance improvement over baseline methods, which supports their claims. The authors also provided sufficient ablation analysis to understand the effective components of the proposed method, showing how the hyper parameters can affect performance.

**Essential References Not Discussed:**

VAE-based augmentation method is classical and has been widely investigated, and the authors can provide slightly more comprehensive literature search to include more related work. Yet the current references are sufficient for reader to understand the presented work.

**Experimental Designs Or Analyses:**

The authors evaluated their method systematically on the UCR Benchmark. The authors also carefully selected the baseline methods, and provided systematic evaluation of the baseline methods on all datasets. It is indeed a common problem that data augmentation methods provide varying performance boost/degradation on diverse set of datasets, and the authors provided experiments to look at this problem carefully, which makes the provided results sound.

One issue is that the ablation experiments are conducted on very specific datasets. Can the authors provide additional rationale on using certain datasets (instead of using other ones) for ablation analysis?

**Methods And Evaluation Criteria:**

The proposed method, especially the iterative training of VAE and sampling of its latent space makes a lot of sense for UCR datasets, as such datasets are often of small scale. The authors evaluated their method on all 102 UCR datasets, provided a systematic evaluation of the proposed method.

It would make more sense if the authors can also apply their method on UEA datasets for multivariate time series classification tasks.

**Other Comments Or Suggestions:**

NA

**Other Strengths And Weaknesses:**

NA

**Questions For Authors:**

NA

**Relation To Broader Scientific Literature:**

The authors nicely summarized the broader scientific literature in the related work section.

**Theoretical Claims:**

There is no theoretical claims. All analytical results seem proper to me.

---

> ### Author Rebuttal · Authors · 2025-04-01
>
> We thank the reviewer for taking the time to read our work and offer such constructive feedback. We appreciate the recognition of the systematic nature of our experiments, as well as the suggestions to discuss the application of our method to multivariate data and expanding our ablation analyses.
>
> ## Methods And Evaluation Criteria
> "**C1.1** *It would make more sense if the authors can also apply their method on UEA datasets for multivariate time series classification tasks.*"
>
> We agree with the reviewer that our method can theoretically apply to multivariate time series classification tasks. However, this would require changing the encoder to account for the multiple dimensions, as well as a specific parameter tuning. This is why we focus our extensive study on univariate time series. We will include a sentence discussing the potential extension to multivariate data, as the rebuttal period is (unfortunately) too short to conduct such adaptations and experiments.
>
> ## Experimental Designs Or Analyses
> "**C1.2** *One issue is that the ablation experiments are conducted on very specific datasets. Can the authors provide additional rationale on using certain datasets (instead of using other ones) for ablation analysis?*"
>
> Regarding the study associated with Figures 7 to 16 (impact of the number of iterations on classification performance), we chose -- for conciseness -- to present results for one representative dataset per UCR category, e.g., one from the 6 ECG Signal datasets, one from the 8 Device datasets, etc. Full results are available in the supplementary material at: https://github.com/ASCENSION-PAPER/ASCENSION/tree/main/alpha_study
>
> We believe this subset sufficiently captures the overall trends, which align with the discussion in Appendix C. If the remaining graphs are deemed essential, we are happy to generate and include them in the Supplementary Material (given this requires significant computation time). We thank the reviewer for highlighting this and will add a clarifying note in the revised version for transparency.
>
> ## Essential References Not Discussed
> "**C1.3** *VAE-based augmentation method is classical and has been widely investigated, and the authors can provide slightly more comprehensive literature search to include more related work.*"
>
> In the initial version of the paper, we did include a comparison with VaDE, a seminal VAE-based data augmentation technique. In addition of VaDE, we incorporated a recent method, KoVAE [Na23], in our new experiments (conducted for the rebuttal), as well as two additional diffusion model-based DA methods, Diffusion-TS [Yu24] and ImagenTime [Na24], in order to strengthen the benchmark study. A summary of the new results is provided below, demonstrating that ASCENSION significantly outperforms all the evaluated baselines (incl., KoVAE). Full version of the new experiments (for all datasets) is available at: https://github.com/ASCENSION-PAPER/ASCENSION/tree/main/comparison_results
>
> These new results will be added to the result section (Sec. 4), and methods discussed in the Related Work (Appendix A), which covers the history of VAE-, diffusion model- and GAN-based data augmentation methods. We are happy to discuss additional references that the reviewer would provide.
>
> - ResNet
>
> | Method       | ↑Nb Augmented | ↑Augmented mean acc | Nb Unchanged | Unchanged mean acc | ↓Nb Worsened | ↑Worsened mean acc | Nb Total | ↑Total mean acc |
> |--------------|---------------|---------------------|--------------|--------------------|--------------|--------------------|----------|-----------------|
> | ASCENSION    | **56** | **4.0**|16 | 0.0 |**30** | -**1.7**| 102 | **1.7** |
> | ImagenTime   | 26 | 1.8 |17 | 0.0  |59 | -6.2 | 102 | -3.1 |
> | Diffusion-TS | 30 | 1.3| 6 | 0.0 |66 | -9.2 | 102 | -5.5|
> | KoVAE        | 1 | 0.7 |6 | 0.0 |95 | -30.6| 102 | -28.5|
>
> - FCN
>
> | Method       | ↑Nb Augmented | ↑Augmented mean acc | Nb Unchanged | Unchanged mean acc | ↓Nb Worsened | ↑Worsened mean acc | Nb Total | ↑Total mean acc |
> |--------------|---------------|---------------------|--------------|--------------------|--------------|--------------------|----------|-----------------|
> | ASCENSION    | **50** | **3.0** |13 | 0.0 |**39** | -**1.4** | 102 | **1.0**|
> | ImagenTime   | 25 | 2.8 |13 | 0.0| 64 | -3.0 | 102 | -1.2 |
> | Diffusion-TS |37 | **3.0** | 7 | 0.0|58 | -14.8| 102 | -7.3 |
> | KoVAE        | 3 | 7.1| 2 | 0.0|97 | -32.5 | 102 | -30.7|
>
> [Na23] Naiman, I. et al. (2023). Generative modeling of regular and irregular time series data via Koopman VAEs. arXiv preprint arXiv:2310.02619.
>
> [Na24] Naiman, I., Berman, N. et al. (2024). Utilizing image transforms and diffusion models for generative modeling of short and long time series. Advances in Neural Information Processing Systems, 37, 121699-121730.
>
> [Yu24] Yuan, X., & Qiao, Y. (2024). Diffusion-ts: Interpretable diffusion for general time series generation. arXiv preprint arXiv:2403.01742.

---

### Decision · Program_Chairs · 2025-05-01

**Decision:**

Reject

**Comment:**

This paper proposes using the following algorithm for augmenting data for time-series classification. Create a latent space with a VAE, create a mixture of gaussians for each class y based on where the latent variables of the class are, sample points from the VAE latent space and assign labels based on their probability of being in a class and then decode the latent variable to get a new augmented data sample to append to the training set. The method for augmentation is used to study the degree to which classifiers improve on time series classification which is tested on the UCR Time Series Archive. The reviews for the work were initially mixed. Some reviewers pointed out that the work needs comparison to more recent state of the art time series classification approaches, others pointed out mismatches in the framing of the paper and a lack of clarity (for example on how ASCENSION expands class probability density and why that provides a better mechanism to help improve the accuracy of the classifer). The rebuttal stated "We hypothesize that controlled, progressive class boundary expansion in latent space can boost classification." but does not directly test this hypothesis via exploratory evidence detailing the mechanism of boosting classification. For example, is it that the classifier sees a more diverse set of samples? is it that the samples are closer to the decision boundary and therefore change the classifier in a manner that helps improve generalization? Overall while some reviewers increased their score, there was no consensus among all and this paper was borderline and fell on the side of rejection. I do encourage the authors to resubmit with the additional experiments and improvements to clarity for the paper. In addition to the suggestions pointed out by the reviewers, it may strengthen the paper to also include a comparison to some of the work on foundation models for time series data (which are effectively built with real and synthetic data); at the very minimum a commentary on why methods for data augmentation are necessitated for time-series classification (relative to using an out of the box time series foundation model) would help readers better situate the work in the literature.